# Graph Few-Shot Learning via Adaptive Spectrum Experts and Cross-Set Distribution Calibration

**Yonghao Liu**[1*]**, Yajun Wang**[1*]**, Chunli Guo**[2*]**, Wei Pang**[3]**, Ximing Li**[1,4]**,**
**Fausto Giunchiglia**[5]**, Xiaoyue Feng**[1†]**, Renchu Guan**[1†]

[1]Key Laboratory of Symbolic Computation and Knowledge Engineering of the Ministry
of Education, College of Computer Science and Technology, Jilin University
[2]College of Software, Jilin University
[3]School of Mathematical and Computer Sciences, Heriot-Watt University
[4]RIKEN Center for Advanced Intelligence Project
[5]Department of Information Engineering and Computer Science, University of Trento
`{yonghao20, yajun24, guocl24}@mails.jlu.edu.cn,`
`w.pang@hw.ac.uk, liximing86@gmail.com, fausto.giunchiglia@unitn.it,`
`{fengxy, guanrenchu}@jlu.edu.cn`

## Abstract

Graph few-shot learning has attracted increasing attention due to its ability to rapidly adapt models to new tasks with only limited labeled nodes. Despite the remarkable progress made by existing graph few-shot learning methods, several key limitations remain. First, most current approaches rely on predefined and unified graph filters (*e.g.*, low-pass or high-pass filters) to globally enhance or suppress node frequency signals. Such fixed spectral operations fail to account for the heterogeneity of local topological structures inherent in real-world graphs. Moreover, these methods often assume that the support and query sets are drawn from the same distribution. However, under few-shot conditions, the limited labeled data in the support set may not sufficiently capture the complex distribution of the query set, leading to suboptimal generalization. To address these challenges, we propose **GRACE**, a novel **G**raph few-shot lea**R**ning framework that integrates **A**daptive spectrum experts with **C**ross-s**E**t distribution calibration techniques. Theoretically, the proposed approach enhances model generalization by adapting to both local structural variations and cross-set distribution calibration. Empirically, GRACE consistently outperforms state-of-the-art baselines across a wide range of experimental settings. Our code can be found here.

## 1 Introduction

Graphs, as a fundamental and expressive data structure, are widely employed to model a variety of complex systems in the real world [1, 2], including social networks [3, 4], transportation networks [5, 6], and protein–protein interaction networks [7, 8]. Recently, graph neural networks (GNNs) have emerged as the *de facto* standard for learning on graph-structured data due to their powerful representation capabilities. However, the effectiveness of GNN-based models heavily relies on the availability of a large number of labeled nodes. A major challenge lies in the fact that annotating large-scale datasets is often impractical in real-world scenarios [9]. This process is not only time- and resource-intensive, but also demands extensive domain-specific expertise in certain specialized fields [10, 11]. For example, in the biomedical domain, accurately annotating unknown genes

---

[*]Equal Contribution

[†]Corresponding Author

39th Conference on Neural Information Processing Systems (NeurIPS 2025).

often requires substantial knowledge of molecular biology, which is difficult even for experienced researchers [12]. In such scenarios where labeled data are scarce, these models often suffer from severe overfitting issues [13]. Thus, graph few-shot learning (FSL) has attracted increasing attention as a promising paradigm that enables rapid adaptation to novel tasks using only a small number of labeled samples. Existing graph FSL models typically follow a two-stage paradigm [14–16]. These models first employ the graph encoder to learn low-dimensional embeddings of nodes, and then apply the few-shot learning algorithm to enable rapid generalization to new tasks. While several graph few-shot learning methods have achieved impressive results [17, 18], they still face several critical limitations that hinder their expressivity.

*First*, most existing graph FSL methods are grounded in either the homophily assumption (*i.e.*, nodes with the same label tend to be connected) or the heterophily assumption (*i.e.*, nodes with different labels tend to be connected) [19]. Based on these assumptions, they typically adopt predefined, uniform graph filters such as low-pass or high-pass filters [20]. This one-size-fits-all design implicitly applies global enhancement or suppression to node frequency signals. However, real-world graph data often exhibit significant local topological heterogeneity, where both homophilic and heterophilic connection patterns may coexist across different local regions of the graph [21, 22]. To substantiate our claim, we visualize the local link distribution of nodes in the Cora dataset [23]. As shown in Fig. 1, it is evident that different nodes exhibit diverse local connectivity patterns. Applying a single, globally designed filter—optimized for a specific connectivity assumption—to all nodes can lead to suboptimal performance and may adversely affect nodes whose local structures deviate from the assumed model. This naturally leads to a fundamental question: *Is it possible to develop a method that enables node-specific filtering strategies to better accommodate the diverse local structures present in real-world graphs?*

*Second*, these graph FSL methods implicitly assume that the support and query sets within each task are drawn from the same underlying distribution. However, this assumption is often challenged in real-world scenarios. On the one hand, the limited labeled data in the support set may fail to adequately capture the complex distribution of the query set [24]. On the other hand, the random sampling process during meta-task construction can introduce systematic biases—such as oversampling from dense subgraphs—which in turn leads to performance degradation under distribution shift conditions. The above claims are further supported by Fig. 2, where we visualize the node distributions of randomly sampled support and query sets on the Cora dataset. As shown in Fig. 2, there exists a clear distributional discrepancy between the two sets,

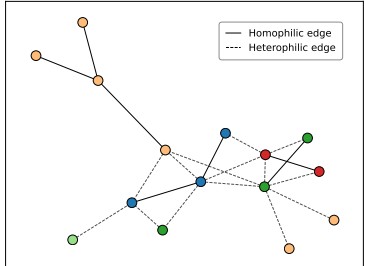

Figure 1: Diversity of local connectivity patterns in the Cora. Node colors indicate their class labels.

highlighting the presence of distribution shift in practical task construction. Hence, effectively narrowing the distribution gap between the support and query sets is essential under distribution shift.

To address the aforementioned challenges, we propose a novel framework named **GRACE**, which integrates both adaptive spectrum experts and cross-set distribution calibration to facilitate effective graph FSL. Specifically, inspired by the mixture-of-experts (MoE) paradigm, we develop a node-specific filtering mechanism that leverages multiple experts to model diverse local connectivity patterns. Each expert is responsible for capturing a distinct graph filtering behavior, while a gating mechanism adaptively assigns expert weights based on the structural characteristics of each node. Next, to alleviate the distributional mismatch between the support and query sets, we initially derive class prototypes from the support set, which are subsequently refined through an explicit calibration process guided by the query set. Theoretically, GRACE enhances the model's generalization lower bound

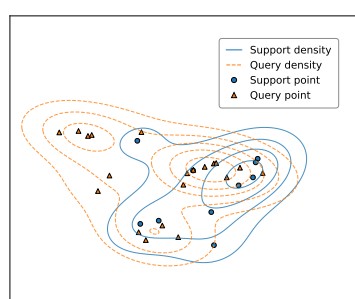

Figure 2: Visualization of distributional discrepancy between support and query sets in the Cora.

by incorporating adaptive spectrum experts that align with local graph structures. Empirically, it

achieves substantial performance gains over competitive baselines on several standard benchmarks. In summary, our contributions are as follows.

(I) We propose a novel framework, GRACE, which integrates adaptive spectrum experts and cross-set distribution calibration to address the challenges of graph FSL.

(II) We provide theoretical analysis showing that GRACE offers improved generalization guarantees by adapting to local structural heterogeneity and mitigating distribution shift.

(III) We conduct extensive experiments on multiple benchmark datasets, demonstrating that GRACE consistently outperforms existing state-of-the-art methods.

## 2 Related Work

**Graph Neural Networks.** GNNs have become the cornerstone in the field of graph-structured data analysis, providing a powerful solution for graph representation learning [1, 25]. Typically, most GNNs follow the message passing mechanism [26], where the nodes continuously aggregate information from their neighboring nodes, gradually extracting local information. This characteristic enables GNNs based on this mechanism to perform excellently when dealing with homophilic graphs. However, when faced with heterophilic graphs, traditional GNNs are clearly inadequate. To this end, researchers have developed a series of specialized models [27, 28]. Recent studies have found that graphs in the real world often exhibit mixed structural patterns [21, 22]. However, traditional GNNs generally adopt a "one-size-fits-all" approach, applying the same global filter to all nodes. This practice cannot fully exploit the characteristics of each node when dealing with graphs with mixed patterns, and it is difficult to achieve the optimal effect. Therefore, our model introduces a node-specific adaptive filtering method, which selects an appropriate filter for each node according to its characteristics.

**Few-Shot Learning.** FSL aims to solve new tasks using a limited number of samples and the knowledge accumulated from previous experiences. This approach has received great attention due to its effectiveness in handling data with rare labels [29–31]. Generally speaking, the existing FSL models can be divided mainly into two categories: (i) optimization-based methods [15, 18, 10] and (ii) metric-based methods [17, 32, 11]. The former focuses on designing different mechanisms to utilize the gradients of samples. For example, the Model-Agnostic Meta-Learning algorithm (MAML) [33] proposes an inner-outer loop mechanism for gradient updates to learn good initial parameters, allowing the model to quickly adapt to new tasks with a small amount of training data. The latter aims to learn a transferable distance metric to evaluate the similarity or degree of association between given samples and query samples. For instance, Prototypical Network [34] calculates the prototype of each category by taking the mean vector of the support examples and classifies query instances by measuring the Euclidean distances between these query instances and the prototypes.

**Mixture-of-Experts.** The MoE architecture [35] is mainly based on the principle of "divide and conquer" [36], that is, first dividing the problem space, and then having specialized sub-models or experts handle their respective parts of the tasks. It has been widely applied in the fields of natural language processing [37, 38] and computer vision [39, 40] to improve the efficiency and performance of large-scale models. Recently, several studies [20, 41–44] in the graph domain have also explored the integration of MoE architectures to enhance graph representation learning. For example, GMoE [42] uses the MoE architecture to adaptively select the propagation hops for different nodes. According to the features of nodes and the information of neighboring nodes, it selects the most suitable propagation hops for each node through a gating mechanism. GraphMETRO [44] utilizes the MoE architecture to address the problem of graph distribution shift. Despite recent progress in applying MoE architectures to general graph learning tasks, their potential remains unexplored in graph FSL scenarios.

## 3 Preliminary Study

In this section, we formally define the studied problem in this work. We focus on few-shot node classification (FSNC), one of the most representative tasks in graph FSL, to evaluate the performance of our proposed model. Formally, we consider an input graph $\mathcal{G} = \{\mathcal{V}, \mathcal{E}, \mathbf{X}, \mathbf{A}\}$, where $\mathcal{V}$ and $\mathcal{E}$ denote the sets of nodes and edges, respectively; $\mathbf{X} \in \mathbb{R}^{n \times d}$ is the node feature matrix; and

$\mathbf{A} \in \{0,1\}^{n \times n}$ is the adjacency matrix, where $\mathbf{A}_{ij} = 1$ if there is an edge between node $i$ and node $j$, and $\mathbf{A}_{ij} = 0$ otherwise. Typically, FSNC consists of two stages: meta-training and meta-testing. The label space of meta-training is denoted as $\mathcal{Y}_{\text{base}}$, and that of meta-testing as $\mathcal{Y}_{\text{new}}$, where $\mathcal{Y}_{\text{base}} \cup \mathcal{Y}_{\text{new}} = \mathcal{Y}$ and $\mathcal{Y}_{\text{base}} \cap \mathcal{Y}_{\text{new}} = \emptyset$. Moreover, we adopt the episodic training paradigm widely used in FSL by constructing a series of meta-tasks. In both in the meta-training and meta-testing phases, the construction of each meta-task follows a consistent procedure. Specifically, each meta-task consists of a support set and a query set, i.e., $\mathcal{T}_t = \{\mathcal{S}_t, \mathcal{Q}_t\}$. The support set is formed by randomly sampling $N$ classes from a particular label space $\mathcal{Y}_*$, and selecting $K$ labeled nodes per class—yielding an $N$-way $K$-shot classification problem, i.e., $\mathcal{S}_t = \{(\mathbf{X}_{t,i}^s, \mathbf{Y}_{t,i}^s)\}_{i=1}^{N \times K}$. The query set is then constructed by sampling $M$ additional nodes per class from the remaining labeled data of those same $N$ classes, i.e., $\mathcal{Q}_t = \{(\mathbf{X}_{t,i}^q, \mathbf{Y}_{t,i}^q)\}_{i=1}^{N \times M}$. Note that the only difference between meta-training and meta-testing tasks lies in the label space from which classes are sampled: the former samples classes from $\mathcal{Y}_{\text{base}}$, while the latter samples from $\mathcal{Y}_{\text{new}}$. The goal of FSNC is to extract generalizable knowledge from a collection of meta-training tasks $\mathcal{T}_{\text{train}} = \{\mathcal{T}_t\}_{t=1}^T$, such that the model can swiftly adapt to a meta-testing task $\mathcal{T}_{\text{test}} = \{\mathcal{S}_{\text{test}}, \mathcal{Q}_{\text{test}}\}$ by leveraging a small support set $\mathcal{S}_{\text{test}} = \{\mathbf{X}_{\text{test},i}^s, \mathbf{Y}_{\text{test},i}^s\}_{i=1}^{N \times K}$ containing only a few labeled instances per class, and accurately predict labels for unseen nodes in the corresponding query set $\mathcal{Q}_{\text{test}} = \{\mathbf{X}_{\text{test},i}^q, \mathbf{Y}_{\text{test},i}^q\}_{i=1}^{N \times M}$.

## 4 Method

In this section, we provide detailed descriptions of our proposed model, GRACE, which consists of two key components: *adaptive spectrum experts* and *cross-set distribution calibration*. The former dynamically assigns expert weights for each node based on its local connectivity patterns, enabling the model to learn more discriminative node embeddings. The latter leverages class prototypes to explicitly calibrate the distribution shift between the support and query sets, thereby enhancing the model's generalization across tasks. To facilitate the better understanding of our model, we illustrate the overall framework of GRACE in Fig. 3.

### 4.1 Adaptive Spectrum Expert

Generally, the first step of FSNC is to learn expressive node embeddings. As previously discussed, existing graph FSL models adopt a fixed graph filter and fail to consider the diverse local connectivity patterns of individual nodes. To this end, we introduce an MoE-based architecture designed to adaptively capture different structural patterns across nodes. Given that real-world graph-structured data often exhibit either homophily or heterophily, we instantiate two experts to model these typical connectivity types: one with low-pass filtering characteristics to smooth node features under homophilic settings, and the other with high-pass filtering behavior to emphasize feature differences in heterophilic regions.

#### 4.1.1 The Low-Pass Expert

It is widely recognized that graph convolutional networks (GCNs) [3] function as low-pass filters [45, 46], effectively capturing smooth node signals. Hence, we select GCNs as one of the experts. The core idea of GCNs is to iteratively aggregate information from the target node's neighbors to update its representation. This process can be formally expressed as:

$$\mathbf{H}^{(\ell+1)} = \sigma(\tilde{\mathbf{D}}^{-\frac{1}{2}} \tilde{\mathbf{A}} \tilde{\mathbf{D}}^{-\frac{1}{2}} \mathbf{H}^{(\ell)} \mathbf{W}^{(\ell)}), \tag{1}$$

where $\tilde{\mathbf{A}} = \mathbf{A} + \mathbf{I}$ is the adjacency matrix with added self-loops, and $\tilde{\mathbf{D}}$ is the corresponding degree matrix. $\mathbf{H}^{(\ell)}$ and $\mathbf{W}^{(\ell)}$ denote the node embeddings at layer $\ell$ and the learned weight matrix, respectively. $\mathbf{H}^{(0)} = \mathbf{X}$ is the initialized original node feature when $\ell = 0$. Moreover, $\sigma(\cdot)$ is the non-linear activation function such as ReLU.

Through the low-pass expert, we can obtain the smoothed node representations $\mathbf{H}_{\text{low}} \in \mathbb{R}^{n \times d'}$ that characterize homophilic connectivity patterns.

#### 4.1.2 The High-Pass Expert

The high-pass expert is designed to amplify the feature differences between connected nodes, thus effectively capturing heterophilic structures where nodes with dissimilar labels are more likely to be

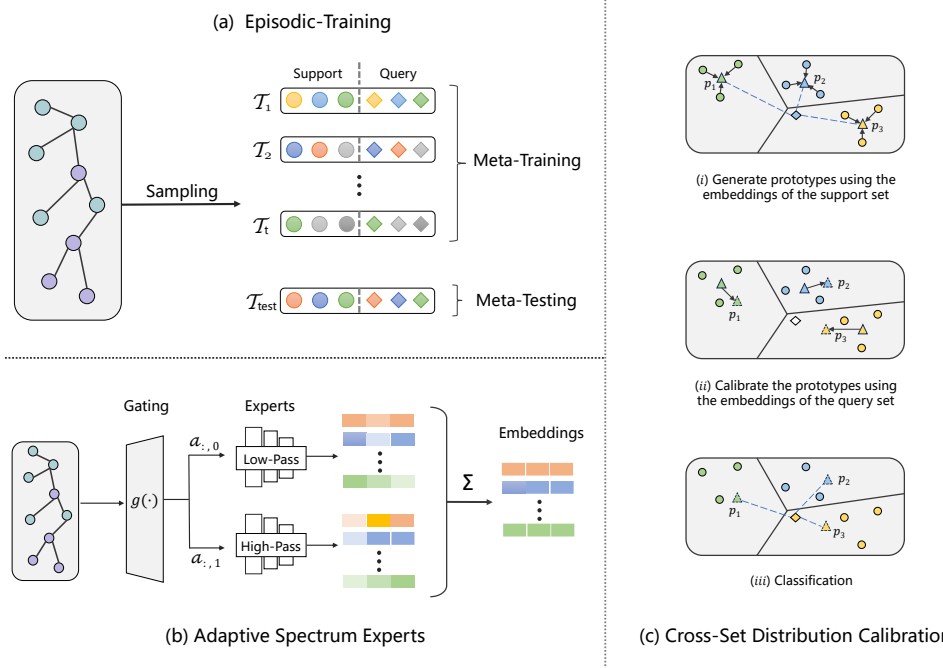

Figure 3: The overall framework of GRACE. (a) Illustration of episodic training. In each episode, an FSNC task is constructed by randomly sampling from the original graph. (b) Adaptive spectrum experts. By introducing multiple experts to capture the diverse frequency components of nodes, we employ a gating module to adaptively weight the spectrum experts. (c) Cross-set distribution calibration. We first compute class prototypes based on the support set. If classification is performed by directly assigning the query sample to the nearest prototype using Euclidean distance, it would be incorrectly assigned to the prototype $p_2$. However, after applying prototype calibration, the query sample can be correctly classified.

linked. To achieve this, we design the following strategy. First, we employ a linear transformation to project the original node features $\mathbf{X}$ into the feature space where smoothed representations $\mathbf{H}_{\text{low}}$ reside. We then compute the difference $\mathbf{F}$ between the original and smoothed features, which explicitly captures the local discrepancy of the node. Finally, we apply an attention mechanism over the resulting differential features, assigning higher weights to neighboring nodes that are significantly different from the target node, thereby enhancing the model's sensitivity to heterophilic connections. The above procedure can be defined as follows:

$$\mathbf{X}' = \mathbf{X}\mathbf{W}', \quad \mathbf{F} = \mathbf{X}' - \mathbf{H}_{\text{low}}, \quad \mathbf{F} = \text{LayerNorm}(\lambda \cdot \mathbf{F}), \tag{2}$$

where $\mathbf{W}' \in \mathbb{R}^{d \times d'}$ is the trainable weight. To ensure stable model training, we apply a scaling factor $\lambda$ to the differential features to control their magnitude, followed by a layer normalization operation.

$$\mathbf{F}_Q = \mathbf{F}\mathbf{W}_Q, \quad \mathbf{F}_K = \mathbf{F}\mathbf{W}_K, \quad \mathbf{F}_V = \mathbf{F}\mathbf{W}_V, \quad \mathbf{H}_{\text{high}} = \text{softmax}(\frac{\mathbf{F}_Q \mathbf{F}_K^\top}{\sqrt{d'}})\mathbf{F}_V, \tag{3}$$

where $\mathbf{W}_Q$, $\mathbf{W}_K$, and $\mathbf{W}_V \in \mathbb{R}^{d' \times d'}$ are the projection matrices. $\mathbf{H}_{\text{high}} \in \mathbb{R}^{n \times d'}$ is the desired high frequency feature, which measures heterophilic connectivity patterns.

### 4.1.3 Gating Module

To effectively integrate the outputs of the low-pass and high-pass experts, we employ a gating mechanism that adaptively assigns weights to each expert's output. Specifically, we concatenate the raw node features $\mathbf{X}$, the absolute difference between the node and its one-hop neighbors $\mathbf{N} = |\hat{\mathbf{A}}\mathbf{X} - \mathbf{X}|$, the feature-wise standard deviation $\phi$ of the raw node features, and the node degree

$\mathbf{D}$ to form the composite representation $\mathbf{X}_g \in \mathbb{R}^{n \times 4d}$, which is then fed into the gating module. This design enables the gating module to dynamically allocate appropriate expert weights based on each node's local topological structure. The procedure can be defined as follows:

$$\mathbf{X}_g = \mathbf{X}||\mathbf{N}||\phi||\mathbf{D}, \quad \alpha = \text{softmax}(\mathbf{X}_g \mathbf{W}_g / \tau), \quad \mathbf{Z} = \alpha_{:,0}\mathbf{H}_{\text{low}} + \alpha_{:,1}\mathbf{H}_{\text{high}}, \tag{4}$$

where $\alpha \in \mathbb{R}^{n \times 2}$ is the gating weights and $\mathbf{W}_g \in \mathbb{R}^{4d \times 2}$ is the trainable weight. $\tau$ is the temperature parameter. $\mathbf{Z} \in \mathbb{R}^{n \times d'}$ is the learned final node embeddings through the adaptive spectrum expert architecture.

### 4.2 Cross-Set Distribution Calibration

Due to the distributional discrepancy between the support and query sets within a meta-task, directly inferring the label of a query node as the nearest prototype in Euclidean space, as done in standard prototypical networks, can easily lead to suboptimal or erroneous decision boundaries. To this end, we propose a cross-set distribution calibration strategy. Specifically, we first compute class prototypes $\mathbf{P} \in \mathbb{R}^{n \times d'}$ based on the support set, *i.e.*, $\mathbf{P}_k = \frac{1}{K}\mathbb{I}[\mathbf{Y}_{t,i} = k]\mathbf{Z}_{t,i}^s$, where $\mathbb{I}[\cdot]$ is the indicator function. Next, inspired by the concept of kernel density estimation (KDE), we calibrate the class prototypes using samples located in high-density regions of the query distribution, which are considered to be more reliable, defined as follows:

$$\Delta = \mathbf{Z}_t^q - \mathbf{P}, \quad \Psi = \text{softmax}(\exp(-\frac{||\Delta||^2}{2\sigma^2})), \tag{5}$$

where $\Delta \in \mathbb{R}^{NM \times d'}$ represents the feature difference between query samples and class prototypes. $\Psi \in \mathbb{R}^{NM \times d'}$ denotes the weight that quantifies the contribution of the query sample to the prototype calibration. $\sigma$ is the bandwidth parameter of the kernel function, which controls the smoothness of the correction process.

Next, we compute a correction vector by performing a weighted summation over the feature differences $\Delta$, using the normalized weights $\Psi$. The calibrated class prototype is then obtained based on this correction vector. The process can be formulated as:

$$\Delta \mathbf{P} = \Delta \odot \Psi, \quad \hat{\mathbf{P}} = \mathbf{P} + \hat{\beta}\Delta\mathbf{P}, \tag{6}$$

where $\hat{\beta} = 0.5(\tanh(\beta) + 1)$ is a trainable parameter that controls the magnitude of prototype calibration, in which $\beta$ is a predefined scalar value.

### 4.3 Model Optimization

After applying the adaptive spectral experts and the cross-set distribution calibration, we adopt the classical metric-based episodic training paradigm for FSNC. Specifically, we optimize the model parameters by minimizing a distance-based cross-entropy loss computed over the query sets of all meta-training tasks in $\mathcal{T}_{\text{train}}$. The objective can be formulated as follows:

$$\mathcal{L} = -\sum_{t=1}^{T}\sum_{i=1}^{NM}\mathbb{I}[\mathbf{Y}_{t,i} = k]\log\frac{\exp(-e(\mathbf{Z}_{t,i}^q\mathbf{W}_l, \hat{\mathbf{P}}_k))}{\sum_{k'}\exp(-e(\mathbf{Z}_{t,i}^q\mathbf{W}_l, \hat{\mathbf{P}}_{k'}))}, \tag{7}$$

where $e(\cdot, \cdot)$ is the Euclidean function and $\mathbf{W}_l$ denotes the trainable vector.

In the meta-testing phase, we also compute the calibrated prototypes using the same strategy as described in Eqs.5 and 6. Then, each query instance is assigned to the class of the nearest calibrated prototype based on Euclidean distance. Formally, the predicted label $\mathbf{Y}_i^q$ for a query instance is defined as follows:

$$\mathbf{P}_k = \frac{1}{|\mathcal{S}_{\text{test},k}|}\sum_{(\mathbf{Z}_{\text{test},i}^s, \mathbf{Y}_{\text{test},i}^s)}\mathbb{I}[\mathbf{Y}_{\text{test},i} = k]\mathbf{Z}_{\text{test},i}^s, \quad \hat{\mathbf{P}} = \text{Calibration}(\mathbf{P}), \quad \mathbf{Y}_{\text{test},*}^q = \text{argmin}_k e(\mathbf{Z}\mathbf{W}_l, \mathbf{P}_k). \tag{8}$$

We present the detailed training procedure and the complexity analysis of GRACE in **Appendices** A.1 and A.2.

# 5 Theoretical Analysis

In this section, we theoretically analyze the effectiveness of the proposed model. Specifically, we present the following theorem to establish the connection between the model's generalization error and the employed techniques.

**Theorem 5.1.** *Suppose that the loss function $\mathcal{L}$ is $L$-Lipschitz continuous, and for $\epsilon_g > 0$, the gating module satisfies $\mathbb{E}_{v \sim P_{\mathcal{V}}} \left[ |\alpha_v - \mathbb{I}(d_v^{\text{hom}} > d_v^{\text{het}})| \right] \leq \epsilon_g$, where $\alpha_v$ denotes the gating weight for node $v$, and $d_v^{\text{hom}}$, $d_v^{\text{het}}$ measure the degrees of homophily and heterophily, respectively. Furthermore, for $\delta > 0$, we assume that the Wasserstein distance between the support and query distributions satisfies $W_1(P_{\mathcal{S}}, P_{\mathcal{Q}}) \leq \delta$. Then the generalization error $\epsilon_{gen}$ of the proposed model is bounded by:*

$$\epsilon_{gen} \leq C_1 \sqrt{\frac{\log T}{T}} + C_2 \epsilon_g + C_3 \left( \delta + \mathcal{O}(\sigma^2) + \mathcal{O}(|\mathcal{Q}|^{-1/2}) \right), \tag{9}$$

*where $C_1$, $C_2$, and $C_3$ are the constants. $T$ and $|\mathcal{Q}|$ are the number of training tasks and query samples. $\sigma$ is the bandwidth.*

Theorem 5.1 indicates that, given a fixed number of training tasks and query samples, our model achieves a tighter generalization error bound compared to that of standard approaches. This improvement is attributed to the reduction of the discrepancy measures $\epsilon_g$ and $\delta$, which is accomplished through the use of adaptive spectrum experts and the cross-domain distribution calibration strategy.

Moreover, we can derive the following corollary to further illustrate the advantage of the adaptive spectral experts.

**Corollary 5.2.** *When local topology exhibits strong heterogeneity ($\epsilon_g \to 0$), our model achieves strictly better bound $\epsilon_{gen}^{MoE}$ than that of single-filter methods $\epsilon_{gen}^{Sin}$:*

$$\epsilon_{gen}^{MoE} \leq \epsilon_{gen}^{Sin} + \mathcal{O}(|\mathcal{Q}|^{-1/2}). \tag{10}$$

Corollary 5.2 indicates that the generalization error of GRACE is clearly lower than that of models using a single filter. The proofs of Theorem 5.1 and Corollary 5.2 can be found in **Appendix** A.3.

# 6 Experiments

## 6.1 Datasets

To empirically validate the effectiveness of our proposed model, we utilize several widely adopted datasets for FSNC tasks, including **Cora** [23], **CiteSeer** [23], **Amazon-Computer** [47], **Coauthor-CS** [47], **DBLP** [48], **CoraFull** [49], and, a large-scale dataset, **ogbn-arxiv** [50]. The statistics of these datasets are presented in Table 1. We present detailed descriptions of these adopted datasets in **Appendix** A.4.

## 6.2 Baselines

To comprehensively evaluate the effectiveness of the proposed model, we compare it against the following three representative categories of baselines. *Graph embedding methods* contain **Deep-Walk** [51], **node2vec** [52], **GCN** [3], and **SGC** [45]. *Meta-learning methods* include **Prototypical Network** (**ProtoNet**) [34] and **MAML** [33]. *Graph meta-learning methods* consist of **GPN** [17], **G-Meta** [18], **TENT** [16], **Meta-GPS** [15], **TEG** [53], **COSMIC** [54], and **Meta-BP** [55]. Detailed descriptions of these baselines are presented in **Appendix** A.5.

Table 1: Statistics of the evaluated datasets.

| Dataset | #Nodes | #Edges | #Features | #Labels |
|---|---|---|---|---|
| Cora | 2,708 | 5,278 | 1,433 | 7 |
| CiteSeer | 3,327 | 4,552 | 3,703 | 6 |
| Amazon-Computer | 13,381 | 245,778 | 767 | 10 |
| Coauthor-CS | 18,333 | 81,894 | 6,805 | 15 |
| DBLP | 40,672 | 144,135 | 7,202 | 137 |
| CoraFull | 19,793 | 65,311 | 8,710 | 70 |
| ogbn-arxiv | 169,343 | 1,166,243 | 128 | 40 |

## 6.3 Implementation Details

In the stage of *adaptive spectrum experts*, the low-pass expert is implemented using a two-layer GCN, with each layer followed by batch normalization and a ReLU activation. For the high-pass

expert, the dimensions of all projection matrices are uniformly set to 32, *i.e.*, $d' = 32$. The gating network is implemented as a two-layer fully connected network, with the hidden dimension of 96. The temperature $\tau$ in Eq.4 is set to 2. Additionally, in the *cross-set distribution calibration* stage, the Gaussian kernel bandwidth $\sigma$ is set to 1. During training, we use the Adam optimizer [56] with an initial learning rate of 0.001. For evaluation, we randomly generate multiple meta-testing tasks from the test set. Specifically, 100 tasks are sampled per evaluation, with each task comprising 10 query samples. To ensure the fairness and stability of our results, we conduct 5 independent experiments and report the average accuracy, standard deviation, and 95% confidence interval across these runs. All experiments are carried out on an NVIDIA 3090Ti GPU to maintain consistent computational conditions and reproducibility.

## 7    Results

**Model Performance.** We conduct extensive experiments across various few-shot settings on multiple datasets. As shown in Tables 2, 3, and 4, our proposed model GRACE consistently achieves superior performance under all experimental configurations, demonstrating its effectiveness for graph FSL compared to other competitive baselines. We attribute the performance improvements to the two key components introduced in our model. The adaptive spectrum experts module leverages an MoE architecture to assign different weights to high-pass and low-pass experts based on the local topology of each target node. This design mitigates the limitations of using a single graph filter, which may fail to accommodate diverse structural patterns. Moreover, the cross-set distribution calibration module leverages the distributional characteristics of high-density samples in the query set to explicitly calibrate the support-set prototypes, effectively narrowing the support–query distribution gap and improving the discriminative power of the classification boundary.

Moreover, it is evident that graph meta-learning models significantly outperform other types of baselines, owing to their tailored designs for addressing the challenges inherent in graph FSL tasks. In contrast, graph embedding methods and conventional meta-learning models exhibit unsatisfactory performance. This performance gap can be attributed to two main reasons: the former lacks mechanisms to cope with the scarcity of labeled nodes and is prone to overfitting, while the latter fails to exploit the inherent structural information of graphs.

Table 2: Accuracies (%) of different models on the three datasets.

| Model | Cora | | | CiteSeer | | | Amazon-Computer | | |
|---|---|---|---|---|---|---|---|---|---|
| | 2 way 1 shot | 2 way 3 shot | 2 way 5 shot | 2 way 1 shot | 2 way 3 shot | 2 way 5 shot | 2 way 1 shot | 2 way 3 shot | 2 way 5 shot |
| DeepWalk | 32.95±2.70 | 36.70±2.99 | 41.51±2.70 | 39.56±2.79 | 39.72±3.42 | 43.22±3.19 | 46.49±2.35 | 49.29±2.46 | 51.24±2.72 |
| node2vec | 31.17±3.16 | 35.66±2.79 | 40.69±2.90 | 40.12±3.15 | 42.39±2.79 | 47.20±2.92 | 49.25±2.56 | 51.46±2.25 | 53.49±2.69 |
| GCN | 55.46±2.16 | 69.96±2.52 | 67.95±2.36 | 51.95±2.45 | 53.79±2.39 | 55.76±2.56 | 60.16±2.20 | 63.46±2.16 | 67.39±2.46 |
| SGC | 56.75±2.31 | 70.15±1.99 | 70.67±2.11 | 53.72±2.55 | 55.12±2.59 | 57.25±2.79 | 61.29±2.45 | 65.39±2.06 | 69.35±2.12 |
| ProtoNet | 50.39±2.52 | 52.67±2.28 | 57.92±2.34 | 49.15±2.29 | 52.19±2.96 | 53.75±2.49 | 57.15±2.55 | 60.49±2.09 | 65.12±2.69 |
| MAML | 52.40±2.29 | 55.07±2.36 | 57.39±2.23 | 49.15±2.25 | 52.75±2.75 | 54.36±2.39 | 53.72±2.25 | 59.20±2.55 | 61.20±2.59 |
| Meta-GNN | 58.82±2.56 | 70.40±2.64 | 72.51±1.91 | 55.45±2.15 | 59.71±2.79 | 61.32±3.22 | 62.36±2.70 | 67.49±2.11 | 70.15±2.16 |
| GPN | 60.12±2.12 | 74.05±1.96 | 76.39±2.33 | 57.36±2.20 | 64.22±2.92 | 65.59±2.49 | 65.56±2.60 | 72.19±2.30 | 76.19±2.21 |
| G-Meta | 59.72±3.15 | 74.39±2.69 | 80.05±1.98 | 54.39±2.19 | 57.59±2.42 | 62.49±2.30 | 64.56±3.10 | 69.49±2.42 | 73.50±2.92 |
| TENT | 55.39±2.16 | 58.25±2.23 | 66.75±2.19 | 60.03±3.11 | 65.20±3.19 | 67.59±2.95 | 80.75±2.95 | 85.32±2.10 | 89.22±2.16 |
| Meta-GPS | 62.19±2.12 | 80.29±2.15 | 83.79±2.10 | 58.95±2.12 | 69.95±2.02 | 72.56±2.06 | 82.12±2.55 | 87.10±2.65 | 90.16±2.05 |
| X-FNC | 61.47±2.99 | 78.19±3.25 | 82.70±3.19 | 58.79±2.56 | 67.96±3.10 | 70.29±3.05 | 81.50±2.29 | 86.39±2.29 | 90.25±2.26 |
| TEG | 62.52±2.95 | 80.65±1.53 | 84.50±2.01 | 59.70±2.69 | 73.79±1.59 | 76.79±2.12 | 86.49±2.10 | 89.02±2.57 | 92.40±2.05 |
| COSMIC | 63.16±2.47 | 65.37±2.49 | 69.10±2.30 | 60.95±2.75 | 70.22±2.56 | 75.10±2.30 | 85.49±2.46 | 88.26±2.02 | 91.59±2.59 |
| TLP | 60.19±2.25 | 71.10±1.66 | 85.15±2.19 | 61.12±2.10 | 71.10±2.17 | 75.55±2.03 | 83.35±2.07 | 89.49±2.06 | 92.09±2.12 |
| Meta-BP | 66.42±4.12 | 76.32±4.30 | 83.12±4.16 | 60.15±2.45 | 72.19±3.19 | 76.11±3.29 | 86.10±4.10 | 89.22±4.29 | 92.39±4.45 |
| **GRACE** | **66.48±2.88** | **82.40±2.03** | **86.19±1.80** | **63.90±2.84** | **75.67±2.44** | **79.64±1.79** | **90.23±0.90** | **92.46±0.55** | **94.66±0.50** |

**Ablation Study.** To validate the effectiveness of the adopted strategies, we design multiple model variants under different few-shot settings on different datasets. (I) *w/o high*: We exclude the high-pass expert. (II) *w/o low*: We discard the low-pass expert. (III) *w/o cal*: We eliminate cross-set distribution calibration. (IV) *w/o both*: We omit both adaptive spectrum expert and cross-set distribution calibration, resulting in a variant that follows the standard training paradigm of graph meta-learning models. We present the ablation results in Table 5, with additional results provided in **Appendix** A.6.

Based on the results, we have the following in-depth analysis. (I) Our proposed GRACE outperforms all four variants, which validates the necessity of the proposed modules. (II) It can be observed that the variant without the cross-set distribution calibration generally exhibits inferior performance.

Table 3: Accuracies (%) of different models on the two datasets.

| Model | Coauthor-CS | | | | DBLP | | | |
|---|---|---|---|---|---|---|---|---|
| | 2 way 3 shot | 2 way 5 shot | 5 way 3 shot | 5 way 5 shot | 5 way 3 shot | 5 way 5 shot | 10 way 3 shot | 10 way 5 shot |
| DeepWalk | 59.52±2.72 | 63.12±3.12 | 33.76±3.21 | 40.15±2.96 | 49.12±2.25 | 59.12±2.32 | 37.11±2.19 | 49.16±2.39 |
| node2vec | 56.16±4.19 | 60.22±4.06 | 30.35±3.93 | 39.16±3.79 | 45.65±2.79 | 55.92±2.36 | 35.72±2.52 | 46.19±2.75 |
| GCN | 73.52±1.97 | 77.20±3.01 | 52.19±2.31 | 56.35±2.99 | 64.12±2.15 | 67.26±2.39 | 42.16±2.39 | 56.12±2.10 |
| SGC | 75.49±2.15 | 79.63±2.01 | 56.39±2.26 | 59.25±2.16 | 66.32±2.25 | 70.19±2.36 | 40.19±2.26 | 55.16±2.56 |
| ProtoNet | 71.18±3.82 | 75.51±3.19 | 47.71±3.92 | 51.66±2.51 | 59.95±2.56 | 62.95±2.72 | 32.35±1.62 | 52.95±1.90 |
| MAML | 62.32±4.60 | 65.20±4.20 | 36.99±4.32 | 42.12±2.43 | 55.05±2.30 | 60.67±2.41 | 29.59±2.90 | 40.22±2.61 |
| Meta-GNN | 85.76±2.74 | 87.86±4.79 | 75.87±3.88 | 68.59±2.59 | 73.41±3.20 | 77.95±3.12 | 65.22±2.79 | 69.12±2.51 |
| GPN | 85.60±2.15 | 88.70±2.21 | 75.88±2.75 | 81.79±3.18 | 75.39±3.41 | 79.90±2.62 | 67.20±2.40 | 71.12±1.87 |
| G-Meta | 92.14±3.90 | 93.90±3.18 | 75.72±3.59 | 74.18±3.29 | 76.49±3.29 | 80.12±2.46 | 68.95±2.70 | 72.19±2.11 |
| TENT | 89.35±4.49 | 90.90±4.24 | 78.38±5.21 | 78.56±4.42 | 78.22±2.10 | 81.30±2.02 | 69.52±2.16 | 73.20±1.95 |
| Meta-GPS | 90.16±2.72 | 92.39±1.66 | 81.39±2.35 | 83.66±1.79 | 79.12±1.92 | 81.66±2.16 | 70.16±2.20 | 73.59±1.26 |
| X-FNC | 90.95±4.29 | 92.03±4.14 | 82.93±2.02 | 84.36±3.49 | 77.45±2.39 | 80.69±2.52 | 69.72±2.39 | 72.95±1.76 |
| TEG | 92.36±1.59 | 93.02±1.24 | 80.78±1.40 | 84.70±1.42 | 79.26±2.49 | 82.19±2.40 | 72.49±2.12 | 73.99±2.55 |
| COSMIC | 89.35±4.49 | 93.32±1.93 | 78.38±5.21 | 85.47±1.42 | 78.34±2.06 | 81.81±2.05 | 66.53±1.54 | 70.09±1.53 |
| TLP | 90.35±4.49 | 90.90±4.24 | 82.30±2.05 | 78.56±4.42 | 77.49±3.22 | 81.95±2.39 | 71.49±2.35 | 73.16±2.30 |
| Meta-BP | 91.19±2.21 | 92.32±2.11 | 81.35±2.02 | 82.12±2.15 | 78.22±2.10 | 81.13±2.55 | 71.30±2.12 | 73.15±2.39 |
| **GRACE** | **95.50±1.30** | **96.20±0.97** | **86.03±1.05** | **86.82±1.01** | **81.72±2.05** | **85.30±1.90** | **74.22±1.56** | **76.70±1.46** |

Table 4: Accuracies (%) of different models on the two datasets.

| Model | CoraFull | | | | ogbn-arxiv | | | |
|---|---|---|---|---|---|---|---|---|
| | 5 way 3 shot | 5 way 5 shot | 10 way 3 shot | 10 way 5 shot | 5 way 3 shot | 5 way 5 shot | 10 way 3 shot | 10 way 5 shot |
| DeepWalk | 23.62±3.99 | 25.93±3.45 | 15.32±4.12 | 17.03±3.73 | 24.12±3.16 | 26.16±2.95 | 20.19±2.35 | 23.76±3.02 |
| node2vec | 23.75±2.93 | 25.42±3.61 | 13.90±3.32 | 15.21±2.64 | 25.29±2.96 | 27.39±2.56 | 22.99±3.15 | 25.95±3.12 |
| GCN | 34.65±2.76 | 39.83±2.49 | 29.23±3.25 | 34.14±2.15 | 32.26±2.11 | 36.29±2.39 | 30.21±1.95 | 33.96±1.59 |
| SGC | 39.56±3.52 | 44.53±2.92 | 35.12±2.71 | 39.53±3.32 | 35.19±2.76 | 39.76±2.95 | 31.99±2.12 | 35.22±2.52 |
| ProtoNet | 33.67±2.51 | 36.53±3.76 | 24.90±2.03 | 27.24±2.67 | 39.99±3.28 | 47.31±1.71 | 32.79±2.22 | 37.19±1.92 |
| MAML | 37.12±3.16 | 47.51±3.09 | 26.61±2.19 | 31.60±2.91 | 28.35±1.49 | 29.09±1.62 | 30.19±2.97 | 36.19±2.29 |
| Meta-GNN | 52.23±2.41 | 59.12±2.36 | 47.21±3.06 | 53.32±3.15 | 40.14±1.94 | 45.52±1.71 | 35.19±1.72 | 39.02±1.99 |
| GPN | 53.24±2.33 | 60.31±2.19 | 50.93±2.30 | 56.21±2.09 | 42.81±2.34 | 50.50±2.13 | 37.36±1.99 | 42.16±2.19 |
| G-Meta | 57.52±3.91 | 62.43±3.11 | 53.92±2.91 | 58.10±3.02 | 40.48±1.70 | 47.16±1.73 | 35.49±2.12 | 40.95±2.70 |
| TENT | 64.80±4.10 | 69.24±4.49 | 51.73±4.34 | 56.00±3.53 | 50.26±1.73 | 61.38±1.72 | 42.19±1.16 | 46.29±1.29 |
| Meta-GPS | 65.19±2.35 | 69.25±2.52 | 61.23±3.11 | 64.22±2.66 | 52.16±2.01 | 62.55±1.95 | 42.96±2.02 | 46.86±2.10 |
| X-FNC | 69.32±3.10 | 71.26±4.19 | 49.63±4.45 | 53.00±3.93 | 52.36±2.75 | 63.19±2.22 | 41.92±2.72 | 46.10±2.16 |
| TEG | 72.14±1.06 | 76.20±1.39 | 61.03±1.13 | 65.56±1.03 | 57.35±1.14 | 62.07±1.72 | 47.41±0.63 | 51.11±0.73 |
| COSMIC | 73.03±1.78 | 77.24±1.52 | 65.79±1.36 | 70.06±1.93 | 52.98±2.19 | 65.42±1.69 | 43.19±2.72 | 47.59±2.19 |
| TLP | 66.32±2.10 | 71.36±4.49 | 51.73±4.34 | 56.00±3.53 | 41.96±2.29 | 52.99±2.05 | 39.42±2.15 | 42.62±2.09 |
| Meta-BP | 72.90±1.90 | 74.36±2.19 | 62.35±2.27 | 67.26±2.59 | 55.12±4.12 | 65.39±4.55 | 46.25±4.52 | 50.12±3.39 |
| **GRACE** | **78.22±1.38** | **81.60±1.28** | **70.91±1.08** | **74.54±0.98** | **62.31±1.94** | **68.34±1.73** | **50.18±1.01** | **55.07±0.91** |

This validates that this strategy can succeed in reducing the distributional discrepancy between the support and query sets within the meta-task. (III) Since real-world graphs often exhibit diverse local connectivity patterns, relying solely on a high-pass or low-pass expert leads to suboptimal performance, highlighting the advantage of GRACE's adaptive combination of both.

Table 5: Results of different model variants on all datasets.

| Model | Cora | CiteSeer | Amazon-Computer | Coauthor-CS | DBLP | CoraFull | ogbn-arxiv |
|---|---|---|---|---|---|---|---|
| | 2 way 1 shot | 2 way 1 shot | 2 way 1 shot | 5 way 3 shot | 5 way 5 shot | 5 way 3 shot | 5 way 3 shot |
| *w/o high* | 64.01±2.67 | 62.26±2.60 | 82.89±2.13 | 79.05±1.23 | 79.35±2.00 | 76.73±1.45 | 59.32±1.90 |
| *w/o low* | 63.94±2.79 | 59.64±2.75 | 90.08±0.79 | 80.31±1.14 | 85.05±1.83 | 77.23±1.49 | 61.98±1.92 |
| *w/o cal* | 65.66±2.80 | 58.61±2.58 | 89.58±1.02 | 85.97±1.13 | 83.52±1.89 | 77.18±1.45 | 61.84±1.96 |
| *w/o both* | 60.12±2.12 | 55.36±2.20 | 65.56±2.60 | 75.88±2.75 | 79.90±2.62 | 53.24±2.33 | 42.81±2.34 |
| Ours | **66.48±2.88** | **63.90±2.84** | **90.23±0.90** | **86.03±1.05** | **85.30±1.90** | **78.22±1.38** | **62.31±1.94** |

**Hyperparameter Sensitivity.** We primarily analyze the impact of two crucial hyperparameters—the bandwidth $\sigma$ and the gating temperature $\tau$—on model performance under the 2 way 5 shot experimental setting across several datasets, as shown in Figs. 4(a) and 4(b). We observe that the model performance with respect to the bandwidth parameter $\sigma$ generally exhibits a rise-then-fall trend. When $\sigma$ is too small, the model only leverages a limited number of neighboring samples. Conversely, an excessively large $\sigma$ introduces many irrelevant or noisy nodes, thereby impairing the model's discriminative capability. Regarding the gating temperature $\tau$, the model performance remains relatively stable across different values. A plausible explanation is that within the explored

range, whether the softmax distribution becomes slightly sharper or smoother, both the low-pass and high-pass experts maintain sufficient representational capacity.

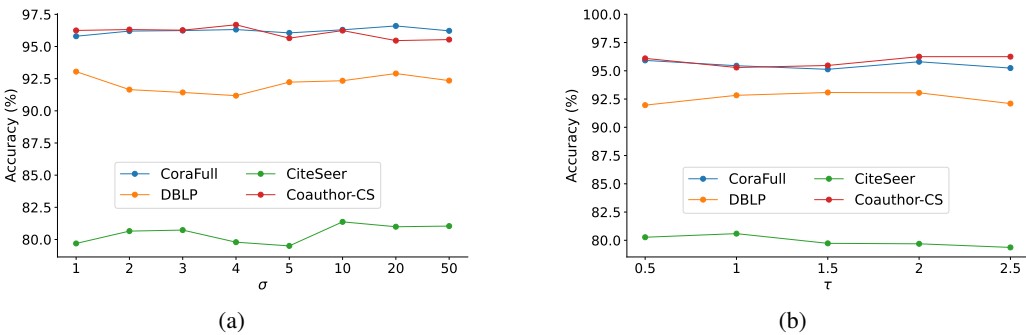

Figure 4: Hyperparameter sensitivity analysis: (a) Model performance varies with the bandwidth $\sigma$. (b) Model performance varies with the gating temperature $\tau$.

**Case Study.** We visualize the weights learned by the gating module to verify whether the proposed adaptive spectral expert strategy can effectively assign different weights to the experts based on varying local connectivity patterns. Specifically, on the Coauthor-CS dataset, nodes are grouped into 20 equal-width bins according to their homophily scores $d_v^{\mathrm{hom}}$. For each bin, we compute and plot the normalized mean weights assigned to the low-pass expert $\alpha_{\mathrm{low}}$ and the high-pass expert $\alpha_{\mathrm{high}}$ (Figs. 5(a) and 5(b)). The results show a clear trend: as node homophily increases, the weight allocated to the low-pass expert steadily increases, while that of the high-pass expert decreases accordingly. This behavior is consistent with our intuition and provides empirical evidence that the proposed adaptive spectral expert strategy effectively captures the local structural patterns of nodes.

Moreover, we visualizes the cross-set distribution calibration strategy under the 2-way 1-shot setting in Fig. 5(c). We observe that the uncalibrated prototypes coincide with the corresponding support samples, which often lie in sparse regions relative to the query samples, potentially leading to misclassification. In contrast, the calibrated prototypes are shifted toward the dense regions of their corresponding query points, bringing them closer to the query cluster centers.

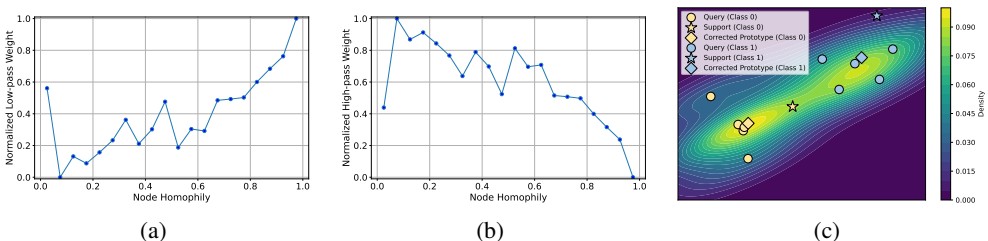

Figure 5: (a) Normalized low-pass expert weight across node homophily $d_v^{\mathrm{hom}}$. (b) Normalized high-pass expert weight across node homophily $d_v^{\mathrm{hom}}$. (c) Cross-set distribution calibration via KDE contours, showing support (stars), query (circles), and corrected prototypes (diamonds).

## 8    Conclusion

In this work, we propose a novel model, named GRACE for graph FSL. Specifically, our model incorporates two key techniques. First, an adaptive spectral expert strategy is employed to assign different weights to multiple experts based on the diverse local connectivity patterns of nodes, thereby learning expressive node embeddings. Second, a cross-set distribution calibration strategy is introduced to mitigate the distributional shift between the support and query sets, enabling the model to establish more accurate decision boundaries. Theoretically, GRACE offers stronger generalization guarantees by adapting to local structural heterogeneity and mitigating distributional shifts. Empirically, GRACE consistently outperforms other competitive models across multiple benchmark datasets.

## Acknowledgments

Our work is supported by the National Science and Technology Major Project under Grant No. 2021ZD0112500, the National Natural Science Foundation of China (No. 62172187 and No. 62372209). Fausto Giunchiglia's work is funded by European Union's Horizon 2020 FET Proactive Project (No.823783).

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

# A  Appendix

## A.1  Training Procedure

---
**Algorithm 1** Training Procedure of GRACE
---
**Require:** A graph $\mathcal{H} = \{\mathcal{V}, \mathcal{E}, \mathbf{X}, \mathbf{A}\}$.
**Ensure:** The trained GRACE.
 1: Perform the low-pass expert using Eq.1.
 2: Perform the high-pass expert using Eqs.2 and 3.
 3: Intergrate the outputs of low-pass and high-pass experts using Eq.4.
 4: Perform cross-set distribution calibration using Eqs.5 and 6.
 5: Optimize the proposed model by minimizing the loss in Eq.7.
 6: Evaluate the model performance using query set with Eq.8.
---

We present the detailed training procedure of GRACE in Algorithm 1.

## A.2  Complexity Analysis

In this section, we provide a detailed analysis of the time complexity of the proposed model. During the graph feature extraction phase, the primary computational cost arises from the low-pass branch and the high-pass expert's forward propagation. The low-pass branch employs two graph convolutional layers implemented using sparse matrix multiplications, resulting in an approximate complexity of $\mathcal{O}(2|E| \cdot d')$, where $|E|$ denotes the number of edges and $d'$ represents the dimensionality of the hidden units. In the high-pass expert module, aside from an initial linear projection, a sparse self-attention computation is performed on all edges, incurring a complexity of $\mathcal{O}(|E| \cdot d')$. Furthermore, the gating network integrates the raw node features with their first- and second-order neighborhood differences via fully-connected mappings, and its forward pass operates with a complexity of $\mathcal{O}(n \cdot d)$, where $n$ is the total number of nodes and $d$ is the input feature dimension. Regarding the few-shot prototype correction mechanism, the complexity is primarily determined by the Euclidean distance computations and subsequent weighted corrections between the support and query samples, which is approximately $\mathcal{O}(N_Q \cdot d')$, with $N_Q$ denoting the number of query samples. Additionally, the supervised contrastive loss employed in the model necessitates concatenating, normalizing, and computing the similarity matrix for the features of both support and query samples, leading to an overall complexity of $\mathcal{O}((N_S + N_Q)^2 \cdot d')$, where $N_S$ represents the number of samples in the support set. In summary, the overall time complexity of the model is chiefly influenced by the number of edges, nodes, and the feature dimensions; by employing sparse matrix operations and localized sampling strategies, we have effectively reduced the computational burden, ensuring that the model's runtime efficiency remains within acceptable bounds for practical applications.

## A.3  Theoretical Proofs

### A.3.1  Proof of Theorem 5.1

Before formally proving Theorem 5.1, we first present several key definitions.

1. **(Lipschitz Continuity)** The loss function $\mathcal{L} : \mathbb{R}^N \times \mathcal{Y} \to \mathbb{R}^+$ is $L$-Lipschitz continuous if the following is satisified:
$$|\mathcal{L}(\mathbf{Z}, y) - \mathcal{L}(\mathbf{Z}', y)| \leq L\|\mathbf{Z} - \mathbf{Z}'\|_2, \quad \forall y \in \mathcal{Y}.$$

2. **(Heterogenity Balance)** For node $v \in \mathcal{V}$, define homophily degree $d_v^{\mathrm{hom}} = |\{(u, v) \in \mathcal{E} : y_u = y_v\}|$ and heterophily degree $d_v^{\mathrm{het}} = |\mathcal{E}_v| - d_v^{\mathrm{hom}}$. For $\epsilon_g > 0$, the gating network satisfies:
$$\mathbb{E}_{v \sim P_\mathcal{V}} \left[|\alpha_v - \mathbb{I}(d_v^{\mathrm{hom}} > d_v^{\mathrm{het}})|\right] \leq \epsilon_g.$$

3. **(Distribution Shift)** For $\delta > 0$, the Wasserstein-1 distance between support set distribution $P_\mathcal{S}$ and query set distribution $P_\mathcal{Q}$ satisfies:
$$W_1(P_\mathcal{S}, P_\mathcal{Q}) \leq \delta.$$

The complete error decomposition is:

$$\epsilon_{gen} = \underbrace{\epsilon_{MoE}}_{\text{Expert Variance}} + \underbrace{\epsilon_{Gate}}_{\text{Gating Error}} + \underbrace{\epsilon_{Dist}}_{\text{Distribution Shift}} . \tag{1}$$

Next, we present the following lemmas to assist the proof of Theorem 5.1.

**Lemma A.1** (Expert Variance Bound). *Let $\mathcal{F}_{MoE}$ be the hypothesis class of the adaptive spetrum experts module. The expected error from expert diversity satisfies:*

$$\epsilon_{MoE} \leq 2L\mathfrak{R}_N(\mathcal{F}_{MoE}) \leq C_1\sqrt{\frac{\log T}{T}},$$

*where $\mathfrak{R}_N$ is the Rademacher complexity.*

*Proof.* The proof follows three key steps:

**Step 1: Define Empirical Rademacher Complexity.** For $T$ *i.i.d.* tasks $\{\mathcal{T}_i\}_{i=1}^T$ and Rademacher variables $\beta_i \in \{\pm 1\}$:

$$\mathfrak{R}_T(\mathcal{F}_{MoE}) = \mathbb{E}\left[\sup_{f \in \mathcal{F}_{MoE}} \frac{1}{T}\sum_{i=1}^N \beta_i f(\mathcal{T}_i)\right].$$

**Step 2: Apply Rademacher Bound for Ensembles.** Using the theorem in [1] for $\Pi$-expert models:

$$\mathfrak{R}_T(\mathcal{F}_{MoE}) \leq \sqrt{\frac{\log T}{T}} \sum_{\pi=1}^\Pi \mathbb{E}\left[\|\mathbf{w}_\pi\|_2\right]$$

$$\leq \sqrt{\frac{\log T}{T}} \cdot \Pi$$

**Step 3: Link to Generalization Error.** By Talagrand's contraction lemma [2]:

$$\mathbb{E}[\mathcal{E}_{MoE}] \leq 2L\mathfrak{R}_T(\mathcal{F}_{MoE})$$

$$\leq 2L\sqrt{\frac{\log T}{T}}$$

$$= C_1\sqrt{\frac{\log T}{T}}.$$

$\square$

**Lemma A.2** (Gating Error Propagation). *Under Definition 2 (Heterogenity Balance), the gating-induced error satisfies:*

$$\epsilon_{Gate} \leq L\sqrt{\mathbb{E}_v\left[(\alpha_v - \alpha_v^*)^2\right]} \leq C_2\epsilon_g.$$

*Proof.* The proof consists of three key phases:

**Step 1: Error Vector Representation.** Define the representation discrepancy between ideal and actual gating:

$$\Delta\mathbf{H}_v = (\alpha_v - \alpha_v^*)\mathbf{H}_{\text{low},v} + (\alpha_v^* - \alpha_v)\mathbf{H}_{\text{high},v}$$

Using the Lipschitz continuity of the loss function:

$$|\mathcal{L}(\mathbf{H}_v) - \mathcal{L}(\mathbf{H}_v^*)| \leq L\|\Delta\mathbf{H}_v\|_2$$

**Step 2: Norm Analysis.** By the filter energy bound $\|\mathbf{H}_{\text{low},v}\|_2, \|\mathbf{H}_{\text{high},v}\|_2 \leq 1$ (normalized representations):

$$\|\Delta\mathbf{H}_v\|_2 \leq |\alpha_v - \alpha_v^*|(\|\mathbf{H}_{\text{low},v}\|_2 + \|\mathbf{H}_{\text{high},v}\|_2) \leq 2|\alpha_v - \alpha_v^*|$$

Taking expectation over nodes:

$$\mathbb{E}_v\|\Delta\mathbf{H}_v\|_2 \leq 2\mathbb{E}_v|\alpha_v - \alpha_v^*| \leq 2\epsilon_g$$

**Step 3: Concentration Inequality.** Applying Cauchy-Schwarz inequality to the loss difference:

$$\epsilon_{\text{Gate}} = \mathbb{E}\left[|\mathcal{L}(\mathbf{H}_v) - \mathcal{L}(\mathbf{H}_v^*)|\right] \leq L\mathbb{E}\|\Delta\mathbf{H}_v\|_2 \leq 2L\epsilon_g = C_2\epsilon_g$$

$\square$

**Lemma A.3** (Distribution Calibration Error). *Let $\hat{P}_{\mathcal{S}}$ be the calibrated support distribution using KDE with bandwidth $\sigma$, and $P_{\mathcal{Q}}$ be the query distribution. Under Definition 3 ($W_1(P_{\mathcal{S}}, P_{\mathcal{Q}}) \leq \delta$), the distribution shift error satisfies:*

$$\epsilon_{Dist} \leq C_3 \left( \delta + \mathcal{O}(\sigma^2) + \mathcal{O}\left(|\mathcal{Q}|^{-1/2}\right) \right)$$

*where $C_3 = L \cdot \mathrm{diam}(\mathcal{Y})$ depends on the label space diameter.*

*Proof.* We analyze the distribution calibration error via three steps:

**Step 1: Error Decomposition.** Using the triangle inequality of Wasserstein distance:

$$W_1(\hat{P}_{\mathcal{S}}, P_{\mathcal{Q}}) \leq \underbrace{W_1(P_{\mathcal{S}}, P_{\mathcal{Q}})}_{\text{Original shift}} + \underbrace{W_1(\hat{P}_{\mathcal{S}}, P_{\mathcal{S}})}_{\text{KDE estimation error}}$$

By definition 3, the first term is bounded by $\delta$.

**Step 2: KDE Estimation Error.** Let $\hat{P}_{\mathcal{S}}(y) = \frac{1}{|\mathcal{Q}|} \sum_{x_j \in \mathcal{Q}} \mathcal{K}_\sigma(y - \tilde{y}_j)$ be the KDE-calibrated distribution, where $\tilde{y}_j$ are perturbed prototypes. Using the Kantorovich-Rubinstein duality [3]:

$$W_1(\hat{P}_{\mathcal{S}}, P_{\mathcal{S}}) = \sup_{\|f\|_L \leq 1} \left| \mathbb{E}_{y \sim \hat{P}_{\mathcal{S}}}[f(y)] - \mathbb{E}_{y \sim P_{\mathcal{S}}}[f(y)] \right|$$

where $f$ is 1-Lipschitz. This can be bounded by:

$$W_1(\hat{P}_{\mathcal{S}}, P_{\mathcal{S}}) \leq \underbrace{\mathbb{E}[|\hat{P}_{\mathcal{S}}(y) - P_{\mathcal{S}}(y)|]}_{\text{Bias}} + \underbrace{\sqrt{\mathrm{Var}(\hat{P}_{\mathcal{S}}(y))}}_{\text{Variance}}$$

**Step 3: Bias-Variance Analysis.** For Gaussian kernel $\mathcal{K}_\sigma$ with bandwidth $\sigma$:

- **Bias term**: By Taylor expansion,

$$\mathbb{E}[\hat{P}_{\mathcal{S}}(y) - P_{\mathcal{S}}(y)] = \mathcal{O}(\sigma^2)$$

- **Variance term**: By the central limit theorem,

$$\mathrm{Var}(\hat{P}_{\mathcal{S}}(y)) = \mathcal{O}\left( \frac{1}{|\mathcal{Q}|\sigma^d} \right)$$

  where $d$ is the feature dimension. Choosing $\sigma \sim |\mathcal{Q}|^{-1/(d+4)}$ optimizes the trade-off:

$$\sqrt{\mathrm{Var}(\hat{P}_{\mathcal{S}}(y))} = \mathcal{O}\left( |\mathcal{Q}|^{-1/2} \right)$$

**Step 4: Final Bound.** Combining all terms with the Lipschitz loss:

$$\mathcal{E}_{Dist} \leq L \cdot \mathrm{diam}(\mathcal{Y}) \cdot W_1(\hat{P}_{\mathcal{S}}, P_{\mathcal{Q}}) \leq C_3 \left( \delta + \mathcal{O}(\sigma^2) + \mathcal{O}\left(|\mathcal{Q}|^{-1/2}\right) \right)$$

where $\mathrm{diam}(\mathcal{Y}) = \sup_{y, y' \in \mathcal{Y}} \|y - y'\|_2$. $\qquad\square$

Next, we formally prove the Theorem 5.1.

*Proof.* Recall Eq.1 and Lemmas A.1, A.2, A.3, the following inequality holds:

$$\epsilon_{gen} = \underbrace{\epsilon_{MoE}}_{\text{Expert Variance}} + \underbrace{\epsilon_{Gate}}_{\text{Gating Error}} + \underbrace{\epsilon_{Dist}}_{\text{Distribution Shift}}$$

$$\leq C_1 \sqrt{\frac{\log T}{T}} + C_2 \epsilon_g + C_3 \left( \delta + \mathcal{O}(\sigma^2) + \mathcal{O}(|\mathcal{Q}|^{-1/2}) \right) \tag{2}$$

$\qquad\square$

Thus, we complete the proof of Theorem 5.1.

### A.3.2 Proof of Corollary 5.2

*Proof.* We compare the generalization error of the proposed model ($\epsilon_{gen}^{MoE}$) with a baseline using a single graph filter ($\epsilon_{gen}^{Sin}$).

**Step 1: Baseline Error Characterization.** For the single-filter baseline, Theorem 5.1 implies:

$$\epsilon_{gen}^{Sin} \leq C_1 \sqrt{\frac{\log T}{T}} + C_2 \epsilon_g^{Sin}.$$

**Step 2: Error Difference Analysis.** Subtract the proposed model's bound (Theorem 5.1) from the baseline:

$$\Delta\epsilon = \epsilon_{gen}^{MoE} - \epsilon_{gen}^{Sin}$$
$$\leq C_2 \left( \epsilon_g - \epsilon_g^{Sin} \right) + \mathcal{O}\left( |\mathcal{Q}|^{-1/2} \right).$$

**Step 3: Gating Advantage.** Under the strong heterogeneity ($\epsilon_g \to 0$), and noting $\epsilon_g^{Sin} \geq \epsilon_g$:

$$C_2(\epsilon_g - \epsilon_g^{Sin}) \leq L(\epsilon_g - \epsilon_g) = 0.$$

**Step 4: Final Inequality.** Combining these results:

$$\Delta\epsilon \leq \mathcal{O}\left( |\mathcal{Q}|^{-1/2} \right).$$

Thus, we complete the proof of Corallary 5.2. $\square$

### A.4 Dataset Descriptions

We conduct experiments on a variety of graph datasets from different domains. Each dataset is divided into disjoint class sets for meta-training, meta-validation, and meta-testing. The details are as follows:

**Cora** [4]: A citation graph where nodes represent academic papers and edges indicate citation relationships. Each node is assigned a label based on the paper's research topic. We divide the dataset into 3, 2, and 2 classes for meta-training, meta-validation, and meta-testing, respectively.

**CiteSeer** [4]: A document-level citation network consisting of scientific publications as nodes and citation links as edges. Labels reflect the thematic area of each document. The dataset is split into 2 classes for each of the three meta-learning phases.

**Amazon-Computer** [5]: A co-purchase network constructed from Amazon product data. Nodes denote products, and edges connect items frequently purchased together. Each product is categorized based on its functional type. We apply a 4/3/3 class split for training, validation, and testing.

**Coauthor-CS** [5]: A collaboration graph in which nodes correspond to authors and edges indicate co-authored publications within the computer science domain. Labels are derived from research specialties. A 5-class split is used for each meta stage.

**DBLP** [6]: A bibliographic co-authorship network where each node denotes a researcher and edges indicate joint publications. Node labels reflect academic fields. We partition the dataset into 77, 30, and 30 classes for training, validation, and testing.

**CoraFull** [7]: An extended version of the Cora dataset that includes a broader range of categories. Nodes represent papers, and citation links define the graph structure. We use 40 classes for meta-training, 15 for validation, and 15 for testing.

**ogbn-arxiv** [8]: A large-scale graph built from arXiv submissions in computer science. Each node corresponds to a paper, and edges are formed based on citation patterns. Labels are based on subject areas defined in the arXiv taxonomy. The dataset is split into 20 classes for training and 10 classes each for validation and testing.

## A.5 Baseline Descriptions

### A.5.1 Graph Embedding Methods

**DeepWalk** [9]: It leverages random walks inspired by the word2vec algorithm to generate low-dimensional node embeddings for graphs.

**GCN** [10]: It employs a first-order Chebyshev approximation graph filter to derive hidden node embeddings, which are then utilized for downstream task analysis.

**SGC** [11]: It streamlines the GCN architecture by eliminating non-linear activations and collapsing weight matrices, resulting in a simpler yet efficient model.

### A.5.2 Meta-Learning Methods

**ProtoNet** [12]: It learns a metric space and predicts query sample categories by measuring their similarity to class prototypes derived from support samples.

**MAML** [13]: By optimizing model parameters through one or few gradient updates, it enables fast adaptation to new tasks with limited labeled data, providing a well-initialized meta-learner.

### A.5.3 Graph Meta-Learning Methods

**GPN** [14]: It adapts ProtoNet by integrating a graph encoder and evaluator to learn node embeddings, assess node importance, and classify new samples based on their proximity to the nearest class prototype.

**G-Meta** [15]: By constructing node-specific subgraphs, it propagates localized node information and employs meta-gradients to extract transferable knowledge across tasks.

**TENT** [16]: It introduces an adaptive framework with node-level, class-level, and task-level components to bridge the generalization gap between meta-training and meta-testing, while minimizing performance fluctuations caused by task variations.

**Meta-GPS** [17]: Enhancing MAML, it incorporates prototype-based parameter initialization, scaling, and shifting transformations to improve meta-knowledge transfer and enable faster adaptation to new tasks.

**TEG** [18]: It designs a task-equivariant graph framework using equivariant neural networks to learn task-adaptive strategies, effectively capturing inductive biases from diverse tasks.

**COSMIC** [19]: It proposes a contrastive meta-learning framework that aligns node embeddings within each episode through a two-step optimization process for improved few-shot learning.

**Meta-BP** [20]: It proposes a lightweight graph meta-learner that extracts relevant knowledge from a black-box pre-trained GNN and leverages task-relevant information to quickly adapt to new tasks, while pruning the meta-learner to enhance its generalization ability on unseen tasks.

## A.6 More Ablation Study

We conduct extensive ablation studies to examine the contribution of individual components in our proposed framework. By systematically removing or altering specific modules, we aim to assess their impact on overall performance and provide insights into the design choices. The detailed results are summarized in Tables A.1, A.2, A.3, and A.4. Specifically, Table A.4 presents an additional ablation on the gating inputs defined in Eq.4, where the input vector is constructed as $\mathbf{X}_g = \mathbf{X} \| \mathbf{N} \| \phi \| \mathbf{D}$, with $\mathbf{X}$ denoting the original node feature, $\mathbf{N} = |\hat{\mathbf{A}}\mathbf{X} - \mathbf{X}|$ the one-hop neighborhood difference, $\phi$ the feature-wise standard deviation, and $\mathbf{D}$ the node degree. We design four variants accordingly: (I) *w/o* $\mathbf{X}$: We remove the original feature; (II) *w/o* $\mathbf{N}$: We discard the neighborhood difference; (III) *w/o* $\phi$: We eliminate the standard deviation; (IV) *w/o* $\mathbf{D}$: We exclude the degree information.

The ablation results clearly demonstrate that each designed module contributes significantly to the overall performance, which is consistent with our analysis in the ablation study section of the main text.

## A.7 Limitation

Although our model achieves outstanding performance in graph few-shot learning, it currently considers only high-pass and low-pass filters. Incorporating a broader range of spectrum experts could potentially further enhance the model's performance. Moreover, it introduces several critical

Table A.1: Results of different model variants on three datasets.

| Model | Cora | | CiteSeer | | Amazon-Computer | |
|---|---|---|---|---|---|---|
| | 2 way 3 shot | 2 way 5 shot | 2 way 3 shot | 2 way 5 shot | 2 way 3 shot | 2 way 5 shot |
| *w/o high* | 74.82±2.49 | 82.89±2.04 | 73.76±2.43 | 77.62±2.03 | 88.34±1.30 | 90.57±1.13 |
| *w/o low* | 78.91±2.11 | 83.37±1.93 | 67.46±2.39 | 70.62±2.23 | 92.06±0.60 | 94.31±5.54 |
| *w/o cal* | 82.35±2.04 | 85.35±1.77 | 71.17±2.44 | 79.32±1.69 | 92.32±0.55 | 94.62±0.52 |
| *w/o both* | 74.05±1.96 | 76.39±2.33 | 64.22±2.92 | 65.59±2.49 | 72.19±2.30 | 76.19±2.21 |
| Ours | **82.40±2.03** | **86.19±1.80** | **75.67±2.44** | **79.64±1.79** | **92.46±0.55** | **94.66±0.50** |

Table A.2: Results of different model variants on two datasets.

| Model | Coauthor-CS | | | DBLP | | |
|---|---|---|---|---|---|---|
| | 2 way 3 shot | 2 way 5 shot | 5 way 5 shot | 5 way 3 shot | 10 way 3 shot | 10 way 5 shot |
| *w/o high* | 93.46±1.41 | 93.00±1.40 | 80.82±1.19 | 76.85±2.11 | 66.81±1.63 | 70.08±1.59 |
| *w/o low* | 94.60±1.34 | 96.18±0.96 | 85.31±1.03 | 79.75±2.03 | 72.50±1.49 | 76.65±1.42 |
| *w/o cal* | 94.98±1.38 | 95.36±1.21 | 86.27±0.95 | 80.14±2.08 | 73.75±1.55 | 75.90±1.49 |
| *w/o both* | 85.60±2.15 | 88.70±2.21 | 81.79±3.18 | 75.39±3.41 | 67.20±2.40 | 71.12±1.87 |
| Ours | **95.50±1.30** | **96.20±0.97** | **86.82±1.01** | **81.72±2.05** | **74.22±1.56** | **76.70±1.46** |

Table A.3: Results of different model variants on two datasets.

| Model | CoraFull | | | ogbn-arxiv | | |
|---|---|---|---|---|---|---|
| | 5 way 5 shot | 10 way 3 shot | 10 way 5 shot | 5 way 5 shot | 10 way 3 shot | 10 way 5 shot |
| *w/o high* | 79.07±1.35 | 67.99±1.13 | 70.55±1.00 | 65.64±1.78 | 47.29±1.01 | 51.55±0.92 |
| *w/o low* | 81.32±1.23 | 67.57±1.15 | 73.95±0.97 | 67.01±1.62 | 49.98±1.04 | 54.94±0.91 |
| *w/o cal* | 80.38±1.30 | 70.09±1.11 | 74.20±0.93 | 67.58±1.75 | 48.46±1.06 | 53.77±0.95 |
| *w/o both* | 60.31±2.19 | 50.93±2.30 | 56.21±2.09 | 50.50±2.13 | 37.36±1.99 | 42.16±2.19 |
| Ours | **81.60±1.28** | **70.91±1.08** | **74.54±0.98** | **68.34±1.73** | **50.18±1.01** | **55.07±0.91** |

Table A.4: Results of different model variants on seven datasets.

| Model | Cora | CiteSeer | Amazon-Computer | Coauthor-CS | DBLP | CoraFull | ogbn-arxiv |
|---|---|---|---|---|---|---|---|
| | 2 way 1 shot | 2 way 1 shot | 2 way 1 shot | 5 way 3 shot | 5 way 5 shot | 5 way 5 shot | 5 way 3 shot |
| *w/o* **X** | 63.46±2.75 | 63.83±2.74 | 89.95±0.80 | 85.36±1.15 | 84.81±1.91 | 80.91±1.30 | 62.46±1.86 |
| *w/o* **N** | 63.12±2.93 | 60.99±2.91 | 89.53±0.84 | 84.63±1.19 | 84.55±1.84 | 80.80±1.27 | 62.06±1.93 |
| *w/o* $\phi$ | 61.46±2.74 | 63.68±3.07 | 89.47±1.01 | 85.39±1.10 | 84.83±1.90 | 81.01±1.20 | 62.19±1.92 |
| *w/o* **D** | 63.80±2.84 | 62.27±2.93 | 89.41±1.03 | 85.26±1.09 | 85.01±1.83 | 81.52±1.28 | 62.26±1.93 |
| Ours | **66.48±2.88** | **63.90±2.84** | **90.23±0.90** | **86.03±1.05** | **85.30±1.90** | **81.60±1.28** | **62.31±1.94** |

hyperparameters that influence the final performance. Determining the optimal settings for these enhancements remains a challenging task. This indicates that there is still room for improvement in our model's performance due to the impact of hyperparameters.

## A.8 Broader Impacts

This study aims to develop an effective approach for graph few-shot learning. Our proposed method not only advances the development of graph-based few-shot learning but may also offer insights for few-shot learning in other domains. While our work does not involve any ethical concerns, it may carry potential societal implications. However, we believe it is not necessary to emphasize them here.

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
