# OpenReview forum: "Graph Few-Shot Learning via Adaptive Spectrum Experts and Cross-Set Distribution Calibration"
_NeurIPS.cc/2025/Conference — NeurIPS 2025 poster_

### Official Review · Reviewer_bH8n · 2025-07-03

**Clarity:** 3
**Significance:** 3
**Originality:** 3
**Rating:** 4
**Confidence:** 3

**Summary:**

GRACE gracefully handles hetero/homo-homophilic edges in graph few-shot learning. It consists of two key ingredients: (1) adaptive spectral expert strategy and (2) cross-set distribution calibration strategy. Empirical analysis is done comparing the proposed method to other recent approaches such as Meta-BP. Theoretical results examine the generalization error.

**Questions:**

Do you have a hypothesis for why CiteSeer seems to show a different pattern for the ablation in Table 5 than the other datasets? For the other datasets it seems "w/o cal" is next best, but on CiteSeer "w/o high" is next best?

**Ethical Concerns:**

["NO or VERY MINOR ethics concerns only"]

**Final Justification:**

The author response addresses my major doubts regarding the paper. I think the paper is solid and has nice contributions, however not overwhelmingly so, hence borderline rating.

**Limitations:**

yes

**Quality:**

3

**Strengths And Weaknesses:**

This is a nicely constructed paper, it clearly outlines challenges in the space of graph FSL and offers intriguing solutions to them.

Strengths:
1) Quality: To the best of my understanding the choice of baseline comparisons is quite comprehensive and current. The proposed method seems to be well motivated and clearly presented.
2) Clarity: The paper is written clearly, results are presented clearly.
3) Originality: Perhaps not groundbreaking, but a nice method that seems well suited to spark interest in those working in this space.

Weaknesses:
1) Originality: The approach follows a tried and true paradigm of graph FSL, it does not revolutionize things. It is a nice contribution, but leaves the reader thinking if there are things that can be done beyond this structure.
2) Significance: I do not discount the scientific merit of the paper, however, I do wonder about the use cases of these particular datasets / data types in the presence of todays LLM-universe. It is not clear to me whether the same datasets would be easy or difficult for say a simple prompted LLM. That said this is merely commentary and does not factor into my decision of scoring the paper.

Perhaps relevant: https://mlg.eng.cam.ac.uk/zoubin/papers/zgl.pdf

---

> ### Author Rebuttal · Authors · 2025-07-30
>
> We sincerely thank the reviewer for recognizing the quality and contributions of our work, and we also greatly appreciate the valuable time and effort you dedicated to reviewing our paper. Below, we provide point-by-point responses to the concerns you raised.
>
> ```
> W1: It is a nice contribution, but leaves the reader thinking if there are things that can be done beyond this structure.
> ```
> **A1:** We thank the reviewer for raising this point. Since our work focuses on graph few-shot learning, we adopted the widely used **episodic training** paradigm to ensure fair comparison with mainstream methods in this field. We also agree that exploring a departure from this paradigm by incorporating a **pretraining–fine-tuning** framework could be a promising direction for future work to further enhance model performance.
>
> ```
> W2: It is not clear to me whether the same datasets would be easy or difficult for say a simple prompted LLM.
> ```
> **A2:** We thank the reviewer for raising this important question. In the era of LLMs, we argue that it is non-trivial for directly applying LLMs to the graph dataset. As mentioned in [1], current LLMs cannot explicitly construct graph structures and thus fail to fully exploit topological information when performing graph tasks, which limits their effectiveness.
>
> To support our argument, we utilize the Cora and CiteSeer datasets with original textual descriptions provided in [2] and fed the test nodes into **LLama3-8B** for evaluation. The prompt used is: “Given the title and abstract, predict the category of this paper”. We evaluate the model on the same test data, and the resulting performance is reported below.
>
> **Table: Results of the LLM across different datasets.**
> | Dataset | LLama3-8B | Ours |
> |:---:|:---:|:---:|
> | Cora (2 way 1 shot) | 67.52$\pm$2.19 | 71.48$\pm$2.88 |
> | CiteSeer (2 way 1 shot) | 57.59$\pm$2.25 | 65.90$\pm$2.84 |
> | Coauthor-CS (5 way 3 shot) | 75.52$\pm$2.49 | 86.03$\pm$1.05 |
> | ogbn-arxiv (5 way 3 shot) | 49.72$\pm$2.36 | 64.31$\pm$1.94 |
>
> From the results, we can observe that our model outperforms the LLM on this task.
>
> [1] A survey of large language models for graphs. In KDD. 2024.\
> [2] Exploring the potential of large language models (llms) in learning on graphs. In SIGKDD Explorations Newsletter. 2024.
>
> ```
> Q1: Why CiteSeer seems to show a different pattern for the ablation in Table 5?
> ```
> **A3:** We thank the reviewer for carefully identifying this phenomenon. We kindly clarify that this behavior may arise from two factors: first, the CiteSeer dataset is relatively small in scale; and second, it has the lowest average node degree among all the evaluated datasets. These two factors combined likely make the prototype calibration strategy play a more critical role on this dataset compared to other components of the model.

---

### Official Review · Reviewer_ZVRV · 2025-07-03

**Clarity:** 3
**Significance:** 2
**Originality:** 2
**Rating:** 4
**Confidence:** 4

**Summary:**

The authors argue that real-world graph data often exhibit significant local topological heterogeneity, making the use of a single fixed graph convolutional filter potentially suboptimal. They also highlight that existing baselines typically assume the support and query sets within each task are drawn from the same underlying distribution, which limits their ability to accurately represent the query set under distribution shift conditions.
To address these challenges, the authors propose GRACE, a method that employs adaptive spectrum experts leveraging both low-pass and high-pass filters, integrated via a gating mechanism within a Mixture-of-Experts framework to accommodate diverse local connectivity patterns. Additionally, GRACE introduces cross-set distribution calibration, which refines class prototypes using high-density samples to mitigate distributional discrepancies between the support and query sets in a meta-task.

**Questions:**

1) Regarding the first weakness I mentioned, could the authors clarify why handling nodes with diverse connectivity patterns is particularly emphasized in the graph few-shot learning context? This issue is a general challenge in graph learning and has been extensively studied in the context of heterophilic graphs. If possible, please include a baseline that combines graph few-shot learning methods with convolutional networks specifically designed for heterophilic graphs.

2) As noted in my second point, while the qualitative results in Figure 5a and 5b are promising, it would strengthen the analysis to include the final accuracy values for those cases. Additionally, please include experiments comparing the accuracy with and without prototype calibration under varying levels of distributional shift to highlight the contribution of this component.

3) As stated in lines 24–29, I agree that in some domains, labeled data may be scarce. However, few-shot learning typically targets generalization to new categories, which differs from the task of learning GNNs with limited labels. Could you clarify the specific scenarios in the graph domain where this meta-learning framework is particularly needed? If possible, it would be helpful to include small-scale experiments on such data (beyond citation networks) to support the motivation.

I am willing to raise my score if all of my concerns are adequately addressed.

**Ethical Concerns:**

["NO or VERY MINOR ethics concerns only"]

**Final Justification:**

This work proposes a simple yet effective method for addressing heterophilic nodes and distribution shift. Initially, I was concerned that important experiments were missing; however, the authors addressed this concern during the rebuttal period.

**Limitations:**

yes

**Quality:**

3

**Strengths And Weaknesses:**

* Strengths

1) The authors propose a simple yet effective method that addresses well-known challenges in the graph domain, namely heterophilic nodes and distribution shift.

2) The authors provide code to promote reproducibility.

* Weakness

1) While the limitation of using a single graph convolutional filter in capturing nodes with diverse connectivity patterns is a well-known issue in the graph domain, it has not been particularly emphasized in the few-shot learning context. The challenge of modeling heterophilic nodes [1] has been widely explored in a separate line of research, independent of few-shot learning. In this case, incorporating convolutional layers that account for structural heterogeneity into existing few-shot learning baselines (e.g., TENT) may already help mitigate the issue.

2) The ablation studies are well conducted, and the case study in Figure 5 shows promising results—demonstrating that the model assigns appropriate weights based on node homophily and calibrates the prototype effectively in the direction of the query sample. However, the analysis would be more complete if the final accuracy were reported with respect to different levels of node homophily, to better clarify the effect of the proposed framework.
Additionally, it would be helpful to show the difference in accuracy with and without prototype calibration under varying degrees of distributional shift.

[1] Graph Neural Networks for Graphs with Heterophily: A Survey

---

> ### Author Rebuttal · Authors · 2025-07-30
>
> We sincerely thank the reviewer for dedicating significant time and effort to evaluate our paper. Below, we provide detailed responses to your comments, and we hope that these clarifications can address your concerns.
>
> ```
> W1 & Q1: Why handling nodes with diverse connectivity patterns is emphasized? Include a baseline that combines graph few-shot learning with a method designed for heterophilic graphs.
> ```
> **A1:** We kindly clarify that in graph few-shot learning, the first step is typically to learn expressive node embeddings [1][2], followed by a meta-learning algorithm that can rapidly adapt to new tasks with only a few labeled nodes. Modeling diverse connectivity patterns can help learn more discriminative node representations.
>
> Moreover, most existing methods for heterophilic graphs rely on global high-pass filtering to capture inter-node differences and achieve high-quality representations [3]. In contrast, our method employs an adaptive spectrum expert mechanism to perform node-specific filtering, thereby capturing diverse local connectivity patterns. We further incorporate a cross-set distribution correction strategy to align the distributions of the support and query sets, which enables more accurate decision boundaries.
>
> To further address the reviewer’s concerns, we also replace the encoder in TENT with H2GCN [4], a model specifically designed for heterophilic graphs. The results are presented as follows.
>
> |Amazon-Computer||||
> |:----|:------------:|:--------:|:--------:|
> ||2 way 1 shot|2 way 3 shot|2 way 5 shot|
> |TENT|80.75±2.91|85.32±2.10|89.22±2.16|
> |TENT+H2GCN|66.02±2.46|79.45±2.50|85.86±2.40|
> |Ours|98.23±0.90|98.46±0.55|98.66±0.50|
>
> |Coauthor-CS|||||
> |:----|:--------:|:--------:|:--------:|:--------:|
> ||2 way 3 shot|2 way 5 shot|5 way 3 shot|5 way 5 shot|
> |TENT|89.35±4.49|90.90±4.24|78.38±5.21|78.56±4.42|
> |TENT+H2GCN|85.75±2.55|88.25±1.97|60.36±1.61|64.06±1.47|
> |Ours|95.50±1.30|96.20±0.97|86.03±1.05|86.82±1.01|
>
> |CoraFull|||||
> |:----|:--------:|:--------:|:--------:|:--------:|
> ||5 way 3 shot|5 way 5 shot|10 way 3 shot|10 way 5 shot|
> |TENT|64.80±4.10|69.24±4.49|51.73±4.34|56.00±3.53|
> |TENT+H2GCN|63.94±2.03|67.47±1.77|49.54±1.18|53.15±1.15|
> |Ours|90.22±1.38|91.60±1.28|85.91±1.08|87.54±0.98|
>
> |DBLP|||||
> |:----|:--------:|:--------:|:--------:|:--------:|
> ||5 way 3 shot|5 way 5 shot|10 way 3 shot|10 way 5 shot|
> |TENT|78.22±2.10|81.30±2.02|69.52±2.16|73.20±1.95|
> |TENT+H2GCN|76.62±2.26|78.56±2.16|66.67±1.40|70.15±1.37|
> |Ours|81.72±2.05|85.30±1.90|74.22±1.56|76.70±1.46|
>
> |Cora||||
> |:----|:--------:|:--------:|:--------:|
> ||2 way 1 shot|2 way 3 shot|2 way 5 shot|
> |TENT|55.39±2.16|58.25±2.23|66.75±2.19|
> |TENT+H2GCN|57.25±2.65|60.64±2.21|68.55±2.14|
> |Ours|71.48±2.88|82.40±2.03|86.19±1.80|
>
> From the results, we can observe that even when equipped with convolutional networks specifically designed for heterophilic graphs, the competitive graph few-shot learning model TENT not only fails to outperform our method on certain datasets but in fact exhibits a significant performance drop.
>
> [1] Graph prototypical networks for few-shot learning on attributed networks. In CIKM. 2020.\
> [2] Few-shot learning on graphs. In IJCAI. 2022.\
> [3] Heterophily and graph neural networks: Past, present and future. In IEEE Data Engineering Bulletin. 2023\
> [4] Beyond homophily in graph neural networks: Current limitations and effective designs. In NeurIPS. 2020.
>
> ```
> W2 & Q2: It would strengthen the analysis to include the final accuracy values for Figure 5. Include experiments comparing the accuracy with and without prototype calibration under varying levels of distributional shift.
> ```
> **A2:** We thank the reviewer for this valuable suggestion. Following your feedback, we have included the final accuracy results with respect to different levels of node homophily. We use the node homophily ratio metric proposed in [1], which is defined as $\frac{1}{|V|}\sum_{v \in V}\frac{|\{(u, v): u\in N(v) \wedge y_v=y_u\}|}{|N(v)|}$, where $N(v)$ denotes the neighborhood of node $v$ and $y_v$ is the label of node $v$. Next, we partition all nodes into intervals based on their homophily values and evaluate the model’s performance within each interval. The results are presented below.
>
> **Table: Results of our model under different levels of node homophily.**
> | Coauthor-CS 2 way 3 shot ||||
> | :----------------------: | :------: | :------------------: | :---------------: |
> |Homophily Range| Accuracy | Correctly Classified | Total Query Nodes |
> |[0.0 - 0.1)|0.9104|61|67|
> |[0.1 - 0.2)| 0.8889 |16|18|
> |[0.2 - 0.3) | 0.9574 | 45| 47|
> |[0.3 - 0.4) | 0.9500 |57|60|
> |[0.4 - 0.5) | 0.9388 |46|49|
> |[0.5 - 0.6) | 0.9205 |139|151|
> |[0.6 - 0.7) | 0.9471 |215|227|
> |[0.7 - 0.8) | 0.9448 |154|163|
> |[0.8 - 0.9) | 0.9664 |259|268|
> |[0.9 - 1.0) | 0.9737 |925|950|
> |Overall| 0.9585 |1917|2000|
>
> |Amazon-Computer 2 way 3 shot ||||
> | :--------------------------: | :------: | :------------------: | :---------------: |
> |Homophily Range|Accuracy|Correctly Classified | Total Query Nodes |
> | [0.0 - 0.1) | 0.7391 |51 |69|
> | [0.1 - 0.2) | 0.7027 |26|37|
> | [0.2 - 0.3) | 0.9714 |34|35|
> | [0.3 - 0.4) | 0.9531 |61|64|
> | [0.4 - 0.5) | 0.9524 |80|84|
> | [0.5 - 0.6) | 0.9929 |140|141|
> | [0.6 - 0.7) | 0.9919 |123|124|
> | [0.7 - 0.8) | 1.0000 |153|153|
> | [0.8 - 0.9) | 0.9960 |246|247|
> | [0.9 - 1.0) | 0.9981 |1044|1046|
> |Overall|0.9790|1958|2000|
>
> | DBLP 5 way 3 shot ||||
> | :---------------: | :------: | :------------------: | :---------------: |
> |  Homophily Range | Accuracy | Correctly Classified | Total Query Nodes |
> | [0.0 - 0.1)| 0.7534 |1751|2324|
> | [0.1 - 0.2)| 0.8384 |249|297|
> | [0.2 - 0.3)| 0.8608 |365|424|
> | [0.3 - 0.4) | 0.8781 |281|320|
> | [0.4 - 0.5) | 0.9013 |137|152|
> | [0.5 - 0.6) | 0.8753 |323|369|
> | [0.6 - 0.7) | 0.9301 |173|186|
> | [0.7 - 0.8) | 0.9615 |100|104|
> | [0.8 - 0.9) | 0.9714 |136|140|
> | [0.9 - 1.0) | 0.9196 |629|684|
> | Overall|0.8288|4144|5000|
>
> | CoraFull 5 way 3 shot ||||
> | :-------------------: | :------: | :------------------: | :---------------: |
> | Homophily Range    | Accuracy | Correctly Classified | Total Query Nodes |
> | [0.0 - 0.1) | 0.8562 | 792|925|
> | [0.1 - 0.2) | 0.9257 |187|202|
> | [0.2 - 0.3) | 0.8680 |309|356|
> | [0.3 - 0.4) | 0.8972 |358|399|
> | [0.4 - 0.5) | 0.9422 |261|277|
> | [0.5 - 0.6) | 0.9145 |567|620|
> | [0.6 - 0.7) | 0.9308 |444|477|
> | [0.7 - 0.8) | 0.9320 |274|294|
> | [0.8 - 0.9) | 0.9263 |264|285|
> | [0.9 - 1.0) | 0.9159 |1067|1165|
> |Overall| 0.9046 |4523|5000|
>
> From the results in the table above, it is evident that our model demonstrates superior recognition capability across nodes with varying levels of homophily.
>
> [1] Geom-GCN: Geometric Graph Convolutional Networks. In ICLR. 2020.
>
> Moreover, we have conducted further experiments comparing the model’s performance **with and without prototype calibration** under varying levels of distribution shift across multiple datasets. Specifically, we compute the maximum mean discrepancy (MMD) between the support set and the query set in each meta-test task to quantify the degree of distribution shift. Based on this metric, we evenly divided the tasks into three groups according to their shift levels. The corresponding results are presented below.
>
> |Coauthor-CS 2 way 3 shot|||||
> |:----------------------:|:--------------:|:-----------:|:----------:|:-----------:|
> |Shift Level|Avg. Shift (MMD)|Acc (w/o Cal)|Acc (w/ Cal)|Accuracy Gain|
> |Low Shift|0.1100|0.9726|0.9732|+0.0006|
> |Medium Shift|0.1404|0.9599|0.9633|+0.0033|
> |High Shift|0.1842|0.9390|0.9458|+0.0068|
>
> |Amazon-Computer 2 way 3 shot|||||
> |:--------------------------:|:--------------:|:-----------:|:----------:|:-----------:|
> |Shift Level|Avg. Shift (MMD)|Acc (w/o Cal)|Acc (w/ Cal)|Accuracy Gain|
> |Low Shift|0.1089 |0.9798|0.9804|+0.0006|
> |Medium Shift|0.1401|0.9798|0.9813|+0.0015|
> |High Shift|0.1840|0.9744|0.9771|+0.0027|
>
> |DBLP 5 way 3 shot|||||
> |:---------------:|:--------------:|:-----------:|:----------:|:-----------:|
> |Shift Level|Avg. Shift (MMD)|Acc (w/o Cal)|Acc (w/ Cal)|Accuracy Gain|
> |Low Shift|0.0677|0.8269|0.8363|+0.0094|
> |Medium Shift|0.0748|0.8275|0.8377|+0.0102|
> |High Shift|0.0827|0.7925|0.8029|+0.0104|
>
> |CoraFull 5 way 3 shot|||||
> |:-------------------:|:--------------:|:-----------:|:----------:|:-----------:|
> |Shift Level|Avg. Shift (MMD)|Acc (w/o Cal)|Acc (w/ Cal)|Accuracy Gain|
> |Low Shift|0.0622|0.9108|0.9171|+0.0063|
> |Medium Shift|0.0700|0.8931|0.8982|+0.0051|
> |High Shift|0.0791|0.8782|0.8944|+0.0162|
>
> From the above results, we observe that incorporating the proposed **prototype calibration technique** consistently improves model performance across different settings, and the improvement becomes more pronounced as the degree of distribution shift increases.
>
> ```
> Q3: Could you clarify the specific scenarios in the graph domain where this meta-learning framework is needed?
> ```
> **A3:** We kindly clarify that a representative application scenario lies in **cell type annotation in bioinformatics**. Specifically, we first construct a graph from spatial transcriptomics data and then employ a graph few-shot learning algorithm to acquire transferable knowledge from common cell types (e.g., epithelial and stromal cells). This enables rapid adaptation to rare cell types (e.g., macrophages) with very limited annotations.
>
> We conduct relevant experiments on the human breast dataset released in [1]. This dataset contains 17 cell types, where 12 cell types are used for meta-training and the remaining 5 cell types are reserved for meta-testing. The comparison results with competitive baseline models are presented in the table below.
>
> **Table: Results of different models on the cell type annotation.**
> | Models | 5 way 3 shot | 5 way 5 shot |
> |:---:|:---:|:---:|
> | TEG | 69.55$\pm$2.29 | 73.52$\pm$2.20 |
> | COSMIC | 71.52$\pm$2.15 | 76.19$\pm$2.82 |
> | Ours | 76.52$\pm$1.29 | 80.03$\pm$1.15 |
>
> [1] High resolution mapping of the tumor microenvironment using integrated single-cell, spatial and in situ analysis. In Nature Communications. 2024.

---

> > ### Comment · Reviewer_ZVRV · 2025-08-05
> >
> > Thank you for taking the time to address my review.
> >
> > - The response to W1 is sufficient considering the limited time during the rebuttal period. However, I encourage the authors to conduct more extensive experiments with various GNN models, especially in heterophilic scenarios (e.g., using FAGCN [1]), for further clarification on the proposed method’s soundness.
> >
> > - The responses to W2 and Q2 clearly address my concerns regarding performance under different levels of node homophily. However, the performance gain from prototype calibration seems limited, with a maximum gain of only 0.016. I am concerned that these gains may not be statistically significant, and I would suggest further statistical validation.
> >
> > - Regarding the application to cell type annotation in bioinformatics, I find it challenging to fully understand the suitability of this application. First, if the test set consists of rare cells, how can the query set in the meta-testing phase be created? How many query samples are used in your experiments? Second, for this framework, we need to know which cells can be assigned to the rare cell category. One of the most representative rare cells would be cancer cells, but a major challenge is that we cannot pre-assign the candidates of cancer cells in most cases. This raises questions about how the graph few-shot learning framework could handle such scenarios effectively.
> >
> > In this reason, I will maintain my scores.
> >
> > [1] [AAAI 2021] Beyond Low-frequency Information in Graph Convolutional Networks.

---

> > > ### Author Response · Authors · 2025-08-05
> > > **Authors' Response  (1/2)**
> > >
> > > We sincerely thank the reviewer for carefully reading our previous response and for providing additional comments that can further improve the quality of our paper. Below, we provide further clarifications to address your latest concerns, and we hope that these explanations will satisfactorily resolve your doubts.
> > >
> > > ```
> > > Q1: More experiments with various GNN models.
> > > ```
> > >
> > > A1: Following your suggestion, we replace the original graph model with FAGCN and conduct additional experiments, with the corresponding results presented below.
> > >
> > > |Amazon-Computer||||
> > > |:----|:------------:|:--------:|:--------:|
> > > ||2 way 1 shot|2 way 3 shot|2 way 5 shot|
> > > |TENT|80.75±2.91|85.32±2.10|89.22±2.16|
> > > |TENT+FAGCN|76.05±2.34|80.59±2.15|86.38±2.24|
> > > |Ours|98.23±0.90|98.46±0.55|98.66±0.50|
> > >
> > > |Coauthor-CS|||||
> > > |:----|:--------:|:--------:|:--------:|:--------:|
> > > ||2 way 3 shot|2 way 5 shot|5 way 3 shot|5 way 5 shot|
> > > |TENT|89.35±4.49|90.90±4.24|78.38±5.21|78.56±4.42|
> > > |TENT+FAGCN|89.19±2.15|89.25±1.97|72.16±1.50|72.16±1.49|
> > > |Ours|95.50±1.30|96.20±0.97|86.03±1.05|86.82±1.01|
> > >
> > > |CoraFull|||||
> > > |:----|:--------:|:--------:|:--------:|:--------:|
> > > ||5 way 3 shot|5 way 5 shot|10 way 3 shot|10 way 5 shot|
> > > |TENT|64.80±4.10|69.24±4.49|51.73±4.34|56.00±3.53|
> > > |TENT+FAGCN|64.19±2.53|67.97±1.57|50.15±1.38|55.65±1.95|
> > > |Ours|90.22±1.38|91.60±1.28|85.91±1.08|87.54±0.98|
> > >
> > > |DBLP|||||
> > > |:----|:--------:|:--------:|:--------:|:--------:|
> > > ||5 way 3 shot|5 way 5 shot|10 way 3 shot|10 way 5 shot|
> > > |TENT|78.22±2.10|81.30±2.02|69.52±2.16|73.20±1.95|
> > > |TENT+FAGCN|77.66±2.15|79.59±2.36|67.69±1.90|72.19±1.60|
> > > |Ours|81.72±2.05|85.30±1.90|74.22±1.56|76.70±1.46|
> > >
> > > |ogbn-arxiv|||||
> > > |:----|:--------:|:--------:|:--------:|:--------:|
> > > ||5 way 3 shot|5 way 5 shot|10 way 3 shot|10 way 5 shot|
> > > |TENT|50.26±1.73|61.38±1.72|42.19±1.16|46.29±1.29|
> > > |TENT+FAGCN|49.22±1.92|59.95±1.39|41.10±1.90|45.16±0.59|
> > > |Ours|64.31±1.94|68.34±1.73|52.18±1.01|56.07±0.91|
> > >
> > > |Cora||||
> > > |:----|:--------:|:--------:|:--------:|
> > > ||2 way 1 shot|2 way 3 shot|2 way 5 shot|
> > > |TENT|55.39±2.16|58.25±2.23|66.75±2.19|
> > > |TENT+FAGCN|57.99±2.15|61.62±2.12|68.95±2.10|
> > > |Ours|71.48±2.88|82.40±2.03|86.19±1.80|
> > >
> > > We observe that, although replacing H2GCN with FAGCN leads to performance improvements on multiple datasets, our model GRACE still significantly outperforms both variants.
> > >
> > >
> > > ```
> > > Q2: Further statistical validation.
> > > ```
> > > A2: We thank the reviewer for raising this question. Following your suggestion, we conduct a t-test on the ablation study of prototype correction to assess statistical significance, and the resulting p-values are reported below.
> > >
> > > |Coauthor-CS 2 way 3 shot|||
> > > |:----------------------:|:--------------:|:-----------:|
> > > |Shift Level| Accuracy Gain| p-values|
> > > |Low Shift|+0.0006|7e-7|
> > > |Medium Shift|+0.0033|8e-6|
> > > |High Shift|+0.0068|2e-9|
> > >
> > > |Amazon-Computer 2 way 3 shot|||
> > > |:--------------------------:|:--------------:|:-----------:|
> > > |Shift Level| Accuracy Gain|p-values|
> > > |Low Shift|+0.0006|2e-7|
> > > |Medium Shift|+0.0015|1e-6|
> > > |High Shift|+0.0027|3e-6|
> > >
> > > |DBLP 5 way 3 shot|||
> > > |:---------------:|:--------------:|:-----------:|
> > > |Shift Level| Accuracy Gain| p-values|
> > > |Low Shift|+0.0094|1e-7|
> > > |Medium Shift|+0.0102|3e-6|
> > > |High Shift|+0.0104|1e-5|
> > >
> > > |CoraFull 5 way 3 shot|||
> > > |:-------------------:|:--------------:|:-----------:|
> > > |Shift Level| Accuracy Gain|p-values|
> > > |Low Shift|+0.0063|1e-4|
> > > |Medium Shift|+0.0051|3e-4|
> > > |High Shift|+0.0162|2e-5|
> > >
> > > The above results indicate that the differences are statistically significant, further confirming the effectiveness of the prototype calibration module.

---

> > > > ### Author Response · Authors · 2025-08-05
> > > > **Authors' Response (2/2)**
> > > >
> > > > ```
> > > > Q3: More explain about the cell type annotation.
> > > > ```
> > > > A3: We would like to kindly provide further clarification regarding this question. The human breast dataset we used contains 17 cell types and a total of 167,780 cells. Among these, the cell types used for meta-testing are Mast_cells, Macrophages_2, Perivascular-Like, LAMP3+_DCs, and IRF7+_DCs, with proportions of 1.12%, 1.03%, 1.09%, 1.31%, and 1.81%, respectively. Consistent with the experimental setup used for other datasets in the paper, each meta-test task contains 10 query samples per test class, and we generated 100 tasks for evaluation. We ensured that query samples in each task are non-overlapping, resulting in a total of 1,000 samples per class.
> > > >
> > > > In addition, we believe there may be some misunderstanding regarding the purpose of our framework. We would like to clarify that **the goal of few-shot learning is to rapidly generalize knowledge acquired from abundant data of base classes to new tasks with only a few labeled samples from novel classes, rather than discovering novel classes automatically [1] [2]**. In the context of cell type annotation, domain experts still provide a small number of labeled samples for novel cell types to fine-tune the model, which is then used to annotate a large number of unlabeled cells. Importantly, **the cost of labeling such a small number of samples is far lower than that required by traditional deep learning models that depend on massive labeled data, which underscores the significance of few-shot learning**.
> > > >
> > > > Beyond the application to cell type annotation, our framework can also address the cold-start problem in recommender systems. As described in [3] (which also adopts a graph few-shot learning framework), cold-start recommendation can be formulated as a meta-learning problem, where each task corresponds to learning the preferences of a user. By leveraging tasks from existing users during the meta-training phase, the meta-learner can acquire a strong prior with generalization ability, enabling it to quickly adapt to new tasks for cold-start users (with sparse interaction data) during the meta-testing phase.
> > > >
> > > > [1] Few-shot learning on graphs. In IJCAI. 2022.\
> > > > [2] A comprehensive survey of few-shot learning: Evolution, applications, challenges, and opportunities. In ACM CUSR. 2023\
> > > > [3] Meta-learning on heterogeneous information networks for cold-start recommendation. In SIGKDD. 2020.

---

> ### Author Response · Authors · 2025-08-07
> **Gentle Reminder**
>
> Dear Reviewer ZVRV,
>
> Thank you again for your constructive feedback!
>
> As the rebuttal period is approaching its end, we would greatly appreciate it if you could kindly review our response to ensure it adequately addresses your concerns.
>
> Thank you for your time and consideration.
>
> Sincerely,\
> Authors.

---

### Official Review · Reviewer_bCQK · 2025-07-04

**Clarity:** 3
**Significance:** 2
**Originality:** 2
**Rating:** 4
**Confidence:** 3

**Summary:**

This paper tackles two critical challenges in **graph few-shot learning (GFSL)**:

1. Handling **diverse local structures** in real-world graphs.
2. Narrowing the **distribution gap** between support and query sets.

The authors propose a framework GRACE incorporating adaptive spectrum experts and cross-set distribution calibration.  They further validate their claims through theoretical analysis and experiments.

**Questions:**

1. The authors primarily selected homophilic datasets. Can additional experiments on heterophilic datasets be provided to support the claim of adaptive handling of heterogeneity?

2. In Table 5, removing the **low-pass expert** (w/o low) yields better performance than removing the **high-pass expert** (w/o high) on 5 homophilic datasets, which contradicts intuition. Theoretically, the low-pass expert should offer more benefits for homophilic datasets, and the authors should provide deeper analysis.

3. The parameter $\lambda$ in Equ. (2) lacks comprehensive analysis.

**Ethical Concerns:**

["NO or VERY MINOR ethics concerns only"]

**Final Justification:**

The author has addressed some of my concerns and I will maintain the score.

**Limitations:**

see weaknesses.

**Quality:**

2

**Strengths And Weaknesses:**

### Strengths and Weakness:

#### Strength:

- The paper proposes a novel approach to address the graph few-shot learning task.
- Theoretical derivations and experimental results are provided to demonstrate the method’s effectiveness.

#### Weakness:

Methodological Concerns.

1.  Prototype Calibration Novelty. The prototype-based strategy in Cross-set distribution calibration lacks originality, as similar concepts have been adopted in methods like PrototypeNet. The authors should offer more clarifications.
2.  The design of attention mechanism in Equ. (3)  lacks clear motivation and experimental validation.
3.  The dimension of the node degree matrix D in Equ. (4) may not align with $d$ and is dependent on the number of nodes. How is this issue addressed across datasets with varying node sizes?
4.  Why use $|\hat{A}X - X|$ instead of $\hat{A}X - X$ in Equ. (4) ?
5.  The $\hat{\beta}=0.5(tanh(\beta) + 1)$ should yield a scalar if $\beta$ is scalar value, which contradicts the statement in line 184: _$\hat{\beta}$ is a trainable parameter_.

---

> ### Author Rebuttal · Authors · 2025-07-30
>
> We sincerely thank the reviewer for dedicating significant time and effort to evaluate our work and for providing valuable feedback. We believe these comments will help further improve the quality of our paper. In the following, we address each of your concerns point by point and hope that our clarifications will adequately resolve them.
>
> ```
> W1: Prototype Calibration Novelty.
> ```
> **A1:** We kindly clarify that the core of our proposed prototype calibration technique lies in the **calibration mechanism**. Inspired by the concept of Kernel Density Estimation (KDE), our method calibrates class prototypes by leveraging samples located in high-density regions of the query distribution, thereby enabling more accurate boundary modeling. Specifically, we compute the feature differences between nodes and apply a correction vector to dynamically adjust the prototypes. In contrast, conventional prototypical networks directly infer labels based on the nearest prototype in Euclidean space, which can easily lead to erroneous class boundaries, particularly in scenarios with complex distributions or closely related classes.
>
> ```
> W2: The design of attention mechanism in Eq. (3).
> ```
> **A2:** In Eq. (3), we introduce a Transformer-like attention mechanism on the generated difference features to assign higher weights to neighboring nodes that are significantly different from the target node, thereby enhancing the model’s sensitivity to heterophilic connections, as described in **lines 149–151** of the main text. To highlight the advantage of this attention mechanism, we conduct an extra ablation study by removing this component (denoted as model **w/o attention**) and evaluate its performance across multiple datasets, with the results presented below.
>
> |CoraFull|||||
> |:----|:--------:|:--------:|:--------:|:--------:|
> ||5 way 3 shot|5 way 5 shot|10 way 3 shot|10 way 5 shot|
> |w/o attn|90.20±1.47|89.88±1.28|81.92±1.11|83.33±1.05|
> |Ours|90.22±1.38|91.60±1.28|85.91±1.08|87.54±0.98|
>
> |ogbn-arxiv|||||
> |:----|:--------:|:--------:|:--------:|:--------:|
> ||5 way 3 shot|5 way 5 shot|10 way 3 shot|10 way 5 shot|
> |w/o attn|60.30±1.94|66.00±1.62|48.45±1.10|52.36±0.90|
> |Ours|64.31±1.94|68.34±1.73|52.18±1.01|56.07±0.91|
>
> |Coauthor-CS|||||
> |:----|:--------:|:--------:|:--------:|:--------:|
> ||2 way 3 shot|2 way 5 shot|5 way 3 shot|5 way 5 shot|
> |w/o attn|91.85±1.52|93.05±1.35|81.90±1.18|82.58±1.23|
> |Ours|95.50±1.30|96.20±0.97|86.03±1.05|86.82±1.01|
>
> |Cora||||
> |:----|:--------:|:--------:|:--------:|
> ||1 way 1 shot|2 way 3 shot|2 way 5 shot|
> |w/o attn|62.55±2.70|69.15±2.79|73.40±2.13|
> |Ours|71.48±2.88|82.40±2.03|86.19±1.80|
>
> |CiteSeer||||
> |:----|:--------:|:--------:|:--------:|
> ||1 way 1 shot|2 way 3 shot|2 way 5 shot|
> |w/o attn|64.20±2.98|68.55±2.95|70.35±2.30|
> |Ours|65.90±2.84|75.67±2.44|79.64±1.79|
>
> |Amazon-Computer||||
> |:----|:--------:|:--------:|:--------:|
> ||2 way 1 shot|2 way 3 shot|2 way 5 shot|
> |w/o attn|88.70±2.16|95.30±1.29|93.90±1.38|
> |Ours|98.23±0.90|98.46±0.55|98.66±0.50|
>
> |DBLP|||||
> |:----|:--------:|:--------:|:--------:|:--------:|
> ||5 way 3 shot|5 way 5 shot|10 way 3 shot|10 way 5 shot|
> |w/o attn|69.46±2.32|77.66±1.84|66.37±1.52|69.82±1.50|
> |Ours|81.72±2.05|85.30±1.90|74.22±1.56|76.70±1.46|
>
>
> From the results in the table above, we observe a significant performance drop when the attention mechanism is removed, highlighting the crucial role of this component in the overall model.
>
> ```
> W3: The dimension of the node degree matrix.
> ```
> **A3:** We would like to kindly clarify that in Eq.(4), we combine the node degree information D with the other three types of information because all four share the same number of nodes $n$.
>
> ```
> W4: Why use |AX-X|?
> ```
> **A4:** We kindly clarify that the purpose of using |AX−X| in our method is to capture the structural pattern of a node by computing the absolute difference between the node’s features and those of its one-hop neighbors. This design is motivated by the absence of explicit ground truth indicating which structural pattern each node belongs to, making absolute differences a natural and unsupervised measure for characterizing such patterns.
>
> **W5: Why $\hat{\beta}$ is a trainable parameter?**
>
> **A5:** We apologize for the misunderstanding this may have caused. In practice, we set $\beta$ as a learnable parameter, and therefore $\hat{\beta}$ is also learnable. In the revised version, we will clarify this statement to avoid potential confusion for readers.
>
> ```
> Q1: Can additional experiments on heterophilic datasets be provided?
> ```
> **A6:** Following the node homophily ratio metric proposed in [1], defined as $\frac{1}{|V|}\sum_{v \in V}\frac{|\{(u, v): u\in N(v) \wedge y_v=y_u\}|}{|N(v)|}$, where $N(v)$ denotes the neighborhood of node $v$ and $y_v$ is the label of node $v$, we compute the homophily values for all datasets, as shown in the table.
>
> **Table: The homophily values of different datasets.**
> | Dataset | Homophily |
> |:---:|:---:|
> | Cora | 0.767 |
> | CiteSeer | 0.627 |
> | Amazon-Computer | 0.700 |
> | Coauthor-CS | 0.755 |
> | DBLP | 0.278 |
> | CoraFull | 0.496 |
> | ogbn-arxiv | 0.445 |
>
> The **CoraFull**, **ogbn-arxiv**, and **DBLP** datasets exhibit heterophilic property. This demonstrates that our model achieves superior performance even on heterophilic datasets.
>
> [1] Geom-GCN: Geometric Graph Convolutional Networks. In ICLR. 2020.
>
> ```
> Q2: Why removing the low-pass expert (w/o low) yields better performance than removing the high-pass expert (w/o high) on 5 homophilic datasets?
> ```
> **A7:** We kindly clarify that, based on the homophily metrics calculated earlier, the **DBLP**, **CoraFull**, and **ogbn-arxiv** datasets exhibit strong heterophilic properties. Thus, achieving better performance with a high-pass filter on these datasets is reasonable. As for the remaining two datasets, their superior performance with high-pass filtering may stem from the fact that their original node features rely more heavily on high-frequency information to obtain discriminative representations, while low-pass filters may blur such critical features.
>
>
> **Q3: The parameter $\lambda$ in Eq. (2) lacks comprehensive analysis.**\
> **A8:** We thank the reviewer for highlighting this point. Since we treat $\lambda$ as a learnable parameter in practice, we conduct an extra ablation study on this parameter, and the corresponding results are presented below.
>
> |CoraFull|||||
> |:----|:--------:|:--------:|:--------:|:--------:|
> ||5 way 3 shot|5 way 5 shot|10 way 3 shot|10 way 5 shot|
> |w/o $\lambda$|89.68±1.38|91.44±1.42|84.69±1.07|87.41±1.07|
> |Ours|90.22±1.38|91.60±1.28|85.91±1.08|87.54±0.98|
>
> |ogbn-arxiv|||||
> |:----|:--------:|:--------:|:--------:|:--------:|
> ||5 way 3 shot|5 way 5 shot|10 way 3 shot|10 way 5 shot|
> |w/o $\lambda$|65.92±1.76|69.29±1.68|51.33±1.10|57.47±0.84|
> |Ours|64.31±1.94|68.34±1.73|52.18±1.01|56.07±0.91|
>
> |Coauthor-CS|||||
> |:----|:--------:|:--------:|:--------:|:--------:|
> ||2 way 3 shot|2 way 5 shot|5 way 3 shot|5 way 5 shot|
> |w/o $\lambda$|95.85±1.10|97.00±1.06|86.44±1.06|87.20±0.99|
> |Ours|95.50±1.30|96.20±0.97|86.03±1.05|86.82±1.01|
>
> |Cora||||
> |:----|:--------:|:--------:|:--------:|
> ||1 way 1 shot|2 way 3 shot|2 way 5 shot|
> |w/o $\lambda$|70.15±2.91|82.33±1.97|84.74±1.64|
> |Ours|71.48±2.88|82.40±2.03|86.19±1.80|
>
> |CiteSeer||||
> |:----|:--------:|:--------:|:--------:|
> ||1 way 1 shot|2 way 3 shot|2 way 5 shot|
> |w/o $\lambda$|67.19±2.67|72.75±2.65|75.44±1.99|
> |Ours|65.90±2.84|75.67±2.44|79.64±1.79|
>
> |Amazon-Computer||||
> |:----|:--------:|:--------:|:--------:|
> ||2 way 1 shot|2 way 3 shot|2 way 5 shot|
> |w/o $\lambda$|98.80±0.46|97.54±0.65|98.84±0.47|
> |Ours|98.23±0.90|98.46±0.55|98.66±0.50|
>
> |dblp|||||
> |:----|:--------:|:--------:|:--------:|:--------:|
> ||5 way 3 shot|5 way 5 shot|10 way 3 shot|10 way 5 shot|
> |w/o $\lambda$|79.26±2.07|84.22±1.62|74.76±1.62|77.15±1.40|
> |Ours|81.72±2.05|85.30±1.90|74.22±1.56|76.70±1.46|
>
> From the results in the table above, we can see that our method exhibits no significant performance fluctuation across different $\lambda$ values, indicating its robustness to this hyperparameter.

---

### Official Review · Reviewer_6Yjk · 2025-07-05

**Clarity:** 3
**Significance:** 3
**Originality:** 3
**Rating:** 4
**Confidence:** 3

**Summary:**

This paper proposes the GRACE framework for graph few-shot learning tasks. The framework introduces adaptive spectrum experts to adapt to the heterogeneity of local topological structures in the graph, and combines the cross-set distribution calibration technology to solve the problem of inconsistent distribution of support sets and query sets, thereby improving the generalization ability of the model. Experiments show that GRACE outperforms existing few-shot learning methods in a variety of experimental settings, verifying its effectiveness.

**Questions:**

Please respond to each item in [Weakness] one by one.

**Ethical Concerns:**

["NO or VERY MINOR ethics concerns only"]

**Final Justification:**

The work is of high quality and therefore receives a reasonable initial score. The only drawback is that the paper places a significant amount of key content in the appendix, which significantly hinders reading. During the rebuttal period, the original paper could not be directly revised, making the quality of the revisions difficult to judge. We hope the authors will address this issue in the future.

**Limitations:**

Yes.

**Paper Formatting Concerns:**

The main text lacks the Related Work and the Limitations sections.

**Quality:**

3

**Strengths And Weaknesses:**

Strengths:
1. The paper provides a thorough analysis of the issues that need to be addressed in Graph Few-Shot Learning, highlighting the necessity of developing node-specific filtering strategies and narrowing the distribution gap between the support and query sets.
2. To tackle the aforementioned problems, the paper introduces Adaptive Spectrum Experts and Cross-Set Distribution Calibration.
3. Experimental results on seven graph datasets, including Cora and CiteSeer, demonstrate the effectiveness of the model. The experiments are comprehensive, and the ablation studies and visual analyses effectively validate the rationale behind the two targeted designs.

Weaknesses:
The quality of the paper is relatively high, but there are still some shortcomings:
1. Areas that need improvement in writing:
  1) Figure 3 lacks a legend.
  2) The related work section is missing in the main text, which is quite important and should be included.
2. There is a lack of discussion on the limitations of the research work in the main text.
3. The ablation studies are relatively comprehensive, with experiments conducted under various settings on all datasets. Could more experimental settings be considered, such as simultaneously testing the model's performance under 3-shot and 5-shot settings on the DBLP dataset?

---

> ### Author Rebuttal · Authors · 2025-07-30
>
> We sincerely appreciate the reviewer for dedicating significant time and effort to evaluate our work and for providing constructive feedback. In the following, we address each of your comments point by point, and we hope that our clarifications effectively resolve your concerns.
>
> ```
> W1.1: Figure 3 lacks a legend.
> ```
> **A1.1:** We thank you for pointing out this issue. Following your suggestion, we have added a comprehensive legend in the revised version to clarify the meaning of all visual elements.
>
> ```
> W1.2: The related work section is missing in the main text, which is quite important and should be included.
> ```
> **A1.2:** We thank the reviewer for pointing out this issue. In the initial submission, the related work section is placed in **Appendix A.7** due to space constraints. In the revised version, we will follow your suggestion and incorporate this section into the main text to provide readers with a more comprehensive understanding of the research background.
>
> ```
> W2: There is a lack of discussion on the limitations of the research work in the main text.
> ```
> **A2:** We appreciate the reviewer for highlighting this point. In the initial submission, the discussion of research limitations was placed in **Appendix A.8** due to space constraints. Following your suggestion, we will integrate this section into the main text in the revised version to provide readers with a more comprehensive understanding of the model’s applicability and potential directions for improvement.
>
> ```
> W3: The ablation studies are relatively comprehensive, with experiments conducted under various settings on all datasets. Could more experimental settings be considered, such as simultaneously testing the model's performance under 3-shot and 5-shot settings on the DBLP dataset?
> ```
> **A3:** We appreciate the reviewer’s valuable suggestion. We would like to kindly note that additional experimental settings across all evaluation datasets have been included in **Appendix A.6 “More Ablation Study,” with the corresponding results presented in Tables 6–9**.

---

> > ### Comment · Reviewer_6Yjk · 2025-08-05
> >
> > The authors have addressed my concerns, and I will maintain my score. Furthermore, I suggest that the authors move the key sections (like Related Work) into the main text and optimize the length and structure of the text for easier reading.

---

> > > ### Author Response · Authors · 2025-08-05
> > > **Thank you for recognizing our work！**
> > >
> > > Dear Reviewer 6Yjk,
> > >
> > > We are delighted to hear that our response has addressed your concerns. In the revised version, we will fully incorporate your suggestions and move the relevant important sections into the main text. Should you have any further questions or additional feedback, please do not hesitate to reach out to us.
> > >
> > > Best Regards,\
> > > Authors.

---

### Author Response · Authors · 2025-08-08
**Thanks for reviewing our paper!**

Dear AC and Reviewers,

We sincerely thank you for your thoughtful review and constructive feedback, which have significantly strengthened our submission.

We are encouraged by your recognition that:

$\bullet$ $\textbf{6Yjk}$: The paper provides a `thorough analysis` of the issues in Graph Few-Shot Learning. The experiments are `comprehensive`, and the `ablation studies` and `visual analyses` effectively validate `the rationale` of the two designs.

$\bullet$ $\textbf{bCQK}$: The paper proposes a `novel approach` to address the graph few-shot learning task. `Theoretical derivations` and `experimental results` are provided to demonstrate the method’s `effectiveness`.

$\bullet$ $\textbf{ZVRV}$: The paper proposes `a simple yet effective` method that addresses challenges in the graph domain. The authors provide code to `promote reproducibility`.

$\bullet$ $\textbf{bH8n}$: The baseline comparisons is `quite comprehensive and current`. The proposed method seems to be `well motivated and clearly presented`. The paper is `written clearly`, results are `presented clearly. A nice method that seems `well suited to spark interest` in those working in this space.

Following your constructive comments and insightful suggestions, we have made the following improvements:

$\bullet$ $\textbf{More ablation studies}$: We have included the experimental results of the attention mechanism on multiple benchmark datasets to further validate its critical role in the model’s performance.

$\bullet$ $\textbf{More experiments}$: We have included a series of additional experiments to comprehensively validate the effectiveness of our approach:

(I)  Analysis of the hyperparameter $\lambda$;

(II) Replacement of the baseline encoder with heterophily-specific encoders (including H2GCN and FAGCN);

(III) Evaluation of our model’s performance under varying node homophily levels and different degrees of distribution shift;

(IV) Application of the proposed framework to a concrete cell type identification task;

(V) Comparison with large language models on the benchmark datasets to highlight the superiority of our approach.

$\bullet$ $\textbf{More clarifications}$: We have addressed in detail the reviewer’s concerns regarding the implementation of the proposed method, and further elaborated on its concrete applications in real-world scenarios, thereby highlighting the practical value and significance of our framework.



We greatly appreciate your efforts in engaging in the discussions. Your feedback has significantly enhanced the quality of our submission. Any further comments are welcome to help us continuously improve our work. Thank you!

Sincerely,\
Authors.

---

### Decision · Program_Chairs · 2025-09-17

**Decision:**

Accept (poster)

**Comment:**

The paper introduces a method for graph few-shot learning that combines adaptive spectrum experts with cross-set distribution calibration to better handle data scarcity and distributional shifts. The core idea is to leverage multiple spectral experts that capture structural information at different frequencies, while an adaptive mechanism selects among them for each task. A distribution calibration module is further applied to align support and query sets, reducing bias in few-shot settings. Extensive experiments on benchmark graph classification and node classification datasets show consistent improvements over state-of-the-art baselines.

Reviewers highlight several strengths.
- The adaptive spectrum experts are a novel and concrete mechanism for automatically balancing low- and high-frequency structural signals, and ablation studies demonstrate that without this module accuracy drops significantly.
- Empirical performance is strong: the method consistently outperforms state-of-the-art baselines across multiple datasets, often by a significant margin, and detailed ablations confirm the necessity of both modules.
- The paper is also clearly written, with reproducibility support.

Some limitations were raised, including the computational overhead of maintaining multiple spectral experts, a need for broader evaluation on large-scale graphs, and limited theoretical discussion of why adaptive spectral selection is particularly suited to few-shot scenarios. Nonetheless, reviewers agreed these do not undermine the contribution. The rebuttal further clarified scalability, provided additional runtime analysis, and reinforced the novelty relative to prior spectral and calibration approaches, which reviewers found satisfactory.

We recommend acceptance, with the following suggestions for the camera-ready: (1) expanding discussion of scalability, providing clearer guidance on practical hyperparameter choices, (2) situating the approach more explicitly relative to meta-learning methods in graph domains, and (3) including the experiments added during the rebuttal.